# Massive entry of BK Polyomavirus induces transient cytoplasmic vacuolization of human renal proximal tubule epithelial cells

Elias Myrvoll Lorentzen[1,2], Stian Henriksen[1,2], Christine Hanssen Rinaldo[1,2]*

1 Department of Microbiology and Infection Control, University Hospital of North Norway, Tromsø, Norway,
2 Metabolic and Renal Research Group, Department of Clinical Medicine, UiT—The Arctic University of Norway, Tromsø, Norway

* christine.rinaldo@unn.no

## Abstract

BK polyomavirus (BKPyV) is a ubiquitous human virus that establishes a persistent infection in renal tubular epithelial cells and mainly causes disease in kidney transplant recipients. The closely related simian polyomavirus SV40 is known to cause cytoplasmic vacuolization in simian kidney cells, possibly increasing progeny release and cell death. This study aimed to determine whether BKPyV causes cytoplasmic vacuolization in primary human renal proximal tubule epithelial cells (RPTECs) and to investigate its potential role in the replication cycle. Using a large infectious dose (MOI 100–1000), a fraction of RPTECs (10–72%) showed early-wave vacuolization from 3 hours post-infection (hpi), which was mainly reversed by 36 hpi. Independent of the infectious dose, late-wave vacuolization occurred around the timepoint of progeny release. BKPyV receptor binding and internalization were required, as neuraminidase pretreatment and preincubation or treatment with a BKPyV-specific neutralizing antibody prevented early or late-occurring vacuolization. Microscopy revealed that the vacuoles were enlarged acidic endo-/lysosomal structures (dextran, EEA1, Rab5, Rab7, LAMP1, and/or Lysoview positive) that contained membrane-bound BKPyV. Time-lapse microscopy and quantitative PCR revealed that cell death and progeny release preceded late-wave vacuolization, mainly affecting cells directly neighboring the lysed cells. Thus, vacuolization had little impact on cell death or progeny release. Addition of the V-ATPase inhibitor Bafilomycin A1 at 0 hpi blocked vacuolization and BKPyV replication, but addition at 2 hpi only blocked vacuolization, suggesting that continuous endosomal acidification and maturation is needed for vacuole formation, but not for BKPyV replication. Our study shows that a massive uptake of BKPyV in RPTECs induces transient enlargement of endo-/lysosomes and is an early event in the viral replication cycle. Vacuolization gives no clear benefit for BKPyV and is possibly the result of a transiently overloaded endocytic pathway. Focal vacuolization around lysed cells suggests that the spread of BKPyV is preferably local.

within the paper and its Supporting Information files.

**Funding:** This work was supported by a grant from the Northern Norway Regional Health Authority - project number HNF1571-21 to CHR. The funders had no role in study design, data collection and analysis, descision to publish, or preparation of the manuscript.

**Competing interests:** The authours have declared that no competing interests exist.

## Author summary

In kidney transplant recipients, high-level BKPyV replication in allograft tubule epithelial cells causes BKPyV nephropathy, a severe infectious complication leading to reduced allograft function and premature allograft loss. In this study, we demonstrate that BKPyV induces cytoplasmic vacuolization in primary RPTECs, and we perform an in-depth characterization of the vacuolization event. We show that vacuolization is an early and transient event in the BKPyV replication cycle that occurs in cells after massive uptake of extrinsically added BKPyV or naturally released progeny virus. Non-infectious BKPyV Vp1-virus-like-particles also caused vacuolization, demonstrating that BKPyV expression is unnecessary. The high-level BKPyV uptake seems to transiently overload the endocytic pathway, which affects cargo transport and digestion. In support of vacuoles being enlarged acidified endo-/lysosomes, the vacuolar H$^+$ ATPase inhibitor bafilomycin A1 prevented vacuolization. Contrasting reports on SV40, inhibitors of the Rac1-JNK signaling pathway did not prevent vacuolization, and vacuolization did not increase host cell lysis and progeny release. Importantly, we demonstrate in real-time that cell lysis preferentially leads to vacuolization of directly adjacent previously uninfected cells. A similar local spread of BKPyV may also play a role *in vivo*, as focal BKPyV replication is typically observed in kidney allograft biopsies from patients with BKPyV nephropathy.

## Introduction

BK Polyomavirus (BKPyV) is among the more than 10 known human polyomaviruses in the *Polyomaviridae* family. It infects up to 95% of the global human population [1–3] and establishes a persistent infection in reno-urinary epithelial cells [4–6]. BKPyV may sporadically replicate and be shed in the urine, but this does not cause disease in healthy individuals. In contrast, in immunosuppressed individuals such as kidney transplant and allogeneic stem cell transplant recipients, unrestricted BKPyV replication can cause BKPyV nephropathy [7] and hemorrhagic cystitis [8], respectively. Moreover, there is accumulating evidence that BKPyV nephropathy may lead to urothelial cancer [9,10]. Regrettably, there is no anti-viral therapy against BKPyV.

BKPyV is a non-enveloped icosahedral virus with a circular double-stranded DNA genome of about 5 kilobases. The viral capsid consists of 72 pentamers of the major capsid protein Vp1 and inside a total of 72 copies of the minor capsid proteins Vp2 and Vp3, connecting the viral genome to Vp1 [11,12]. Simian virus 40 (SV40), a simian polyomavirus with about 70% sequence homology to BKPyV [13], was initially discovered due to its ability to induce widespread cytoplasmic vacuolization [14]. Vacuoles are membrane-bound organelles enriched with hydrolytic enzymes and named by their transparent morphology [15]. They are best known from plant and fungal cells but are also observed in highly differentiated mammalian tissues. The vacuolization in SV40-infected cells appears to depend on the binding of SV40 to GM1 gangliosides on the host cell surface, as it can be prevented by enzymatic removal of GM1, by antibody-mediated neutralization of SV40 and by alteration of the SV40 receptor usage [16,17]. The role of vacuolization in the SV40 replication cycle is only partially understood. A recent study proposed that vacuolization is induced late in SV40 infection due to activation of the RAS-MAPK signaling pathway and that this supports lytic progeny release [18].

The evidence regarding BKPyV and cytoplasmic vacuolization is conflicting. BKPyV infection has been reported to induce cytoplasmic vacuolization in human fetal neural cells [19], human embryonic fibroblasts [20], and simian Vero cells [21,22], but not in simian CV-1 cells

[16]. To our knowledge, vacuolization has never been described in BKPyV-infected primary human renal proximal tubule epithelial cells (RPTECs), cells that have been widely used for BKPyV studies [23–30] since they closely mimic the host cells *in vivo*. However, vacuolization of renal tubular epithelial cells has been detected in kidney biopsies from patients with BKPyV nephropathy [31,32]. Moreover, when studying the dissemination of BKPyV using polarized RPTECs, we recently observed cytoplasmic vacuolization around the time of progeny release [33]. We therefore sought to determine whether BKPyV can induce cytoplasmic vacuolization in RPTECs and to characterize the role, if any, this phenomenon plays in the replication cycle of BKPyV.

## Results

### BKPyV induces massive cytoplasmic vacuolization in RPTECs

To investigate if BKPyV could induce cytoplasmic vacuolization, we infected RPTECs with BKPyV (multiplicity of infection (MOI) 0.1, 1, and 10) and imaged the cells at 96 hours post-infection (hpi). Widefield microscopy demonstrated that several cells contained multiple translucent vacuoles (**Fig 1A**). Of note is that the number of vacuolized cells increased with increasing infectious doses.

As cytoplasmic vacuolization during SV40 infection seems to depend on an interaction between SV40 and the receptor on the plasma membrane [16, 17], we hypothesized that the vacuolization of RPTECs depends on the binding of progeny BKPyV to sialic-acid containing receptors on the plasma membrane. We, therefore, examined if a large infectious dose could induce early vacuolization. We infected RPTECs with BKPyV MOI 100 and performed microscopy at 6, 24, and 48 hpi. Obvious vacuolization at both 6 hpi and 24 hpi was observed, while at 48 hpi, hardly any vacuoles could be seen (**Fig 1B**). Notably, there were no gross cyto-pathic effects, suggesting that the vacuoles were transient. Next, we infected RPTECs with different MOIs (MOI 10–1000) and quantitated vacuolization at 6 hpi by widefield microscopy. We found that BKPyV at a MOI of 100, 200, and 1000 caused vacuolization in 10%, 28%, and 72% of the cells, respectively, while no obvious vacuolization was observed after infection with MOI 10 (**Figs 1C and S1A**). Moreover, since we expected all cells to be infected when a MOI of 10 or more was used, our results demonstrate that only a fraction of infected cells undergo vacuolization. From here on, transient vacuolization present from a few hours to about 24 hours after inoculation of the culture with a high infectious BKPyV dose is denoted early-wave vacuolization, while vacuolization that occurs at the time of progeny release is denoted late-wave vacuolization. Notably, the temporal aspect, early and late, is related to the time of infection or inoculation of the cell culture and not the replication cycle of the individual cells in the cell culture.

As previously mentioned, we recently detected late-wave vacuolization in polarized BKPyV-infected RPTECs [33]. To investigate if polarized RPTECs undergo early vacuolization, we infected polarized RPTECs with BKPyV ~MOI 100 (based on non-polarized RPTECs) and performed widefield microscopy at 24 hpi. This showed numerous vacuolized cells, confirming that BKPyV could induce early-wave cytoplasmic vacuolization in polarized RPTECs as well (**Fig 1D**). Since all experiments had been performed with RPTECs from one donor, we next infected RPTECs from a different commercial source (Sciencell) with BKPyV MOI 5 and 100 (based on RPTECs from Lonza). As before, when using MOI 100, vacuolization was observed at 12 and 96 hpi, while with MOI 5, vacuolization was only observed at 96 hpi (**S1B Fig**). The result demonstrates that vacuolization is not dependent on RPTECs from one particular donor. Lastly, we assessed if BKPyV could cause vacuolization in less permissive cell lines such as HeLa and CV-1 [34]. Using a MOI of 100 based on RPTECs, neither CV-1 cells (**S2A**

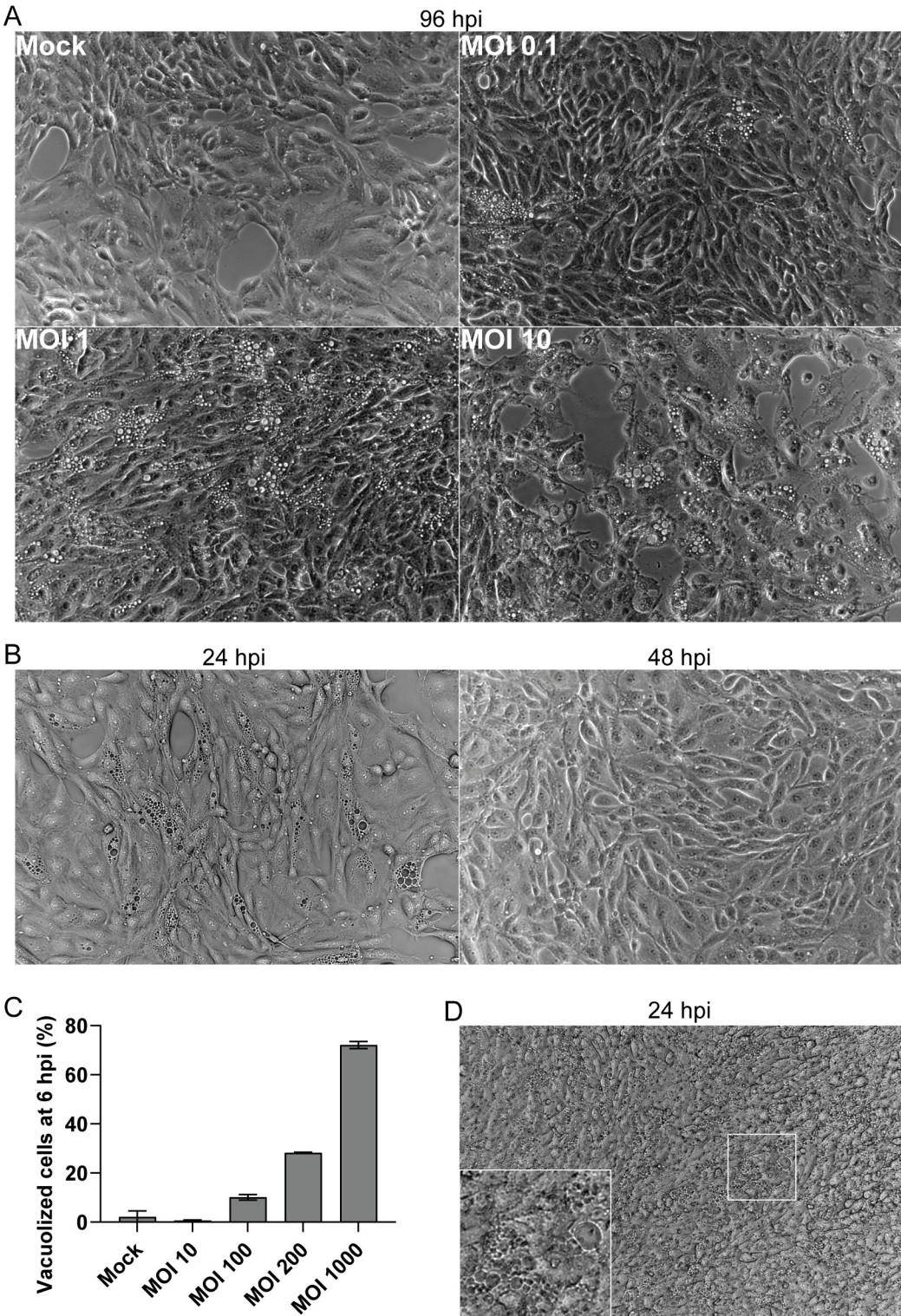

**Fig 1. BKPyV induces cytoplasmic vacuolization in RPTECs. (**A) Phase-contrast images of mock-infected and BKPyV-infected RPTECs (MOI 0.1, 1.0 and 10) at 96 hpi. Representative images from three independent experiments are shown. (B) Phase-contrast images of BKPyV-infected RPTECs (MOI 100) at 24 and 48 hpi, demonstrating the transient appearance of vacuoles. Representative images from three independent experiments are shown. (C) Percentage of vacuolization in mock-infected and BKPyV-infected RPTECs (MOI 10, 100, 200, and 1000) at 6 hpi. The data represents the mean percentage of vacuolized cells based on widefield microscopy with oblique contrast and Hoechst staining. Error

bars represent standard deviation (SD) and n = 2. (D) Phase-contrast image of BKPyV-infected (MOI 100) polarized RPTECs at 24 hpi. A representative image from three independent experiments is shown.

Fig) nor HeLa cells (S2B Fig) showed vacuolization at 24 hpi or 96 hpi. To compare the binding and uptake of BKPyV in HeLa, CV-1, and RPTECs, we infected the cells with BKPyV MOI 200 and performed immunofluorescence staining for Vp1 and Rab7 at 6 hpi. Interestingly, confocal microscopy demonstrated similar levels of Vp1-staining for all three cell types, but again, only RPTECs were vacuolized (S2C Fig).

To summarize, BKPyV induces both transient early-wave vacuolization and late-wave vacuolization in non-polarized and polarized RPTECs but not in HeLa and CV-1 cells.

## The binding of intact viral particles to the plasma membrane is necessary for vacuolization

We next assessed whether heat-denatured viral particles lacking the native conformation could induce vacuolization. We infected RPTECs with heated BKPyV (MOI 1000) and used the same amount of non-heated BKPyV as a positive control. At 5 hpi, widefield microscopy revealed that heat-denatured BKPyV hardly caused any vacuolization compared to the positive control (Fig 2A). This suggests that intact or partially intact viral particles are necessary for the induction of vacuolization.

To investigate if binding of the virus to the viral receptors at the plasma membrane is necessary for vacuolization, we performed two different antibody-mediated neutralization experiments with a mouse monoclonal BKPyV Vp1-specific neutralizing antibody [33]. Like most monoclonal neutralizing antibodies directed against BKPyV [35], this antibody interferes with virus-receptor binding, as shown by its prevention of BKPyV-induced erythrocyte hemagglutination (S3A Fig). First, we preincubated BKPyV (MOI 200) with the BKPyV-specific neutralizing antibody before we infected RPTECs with the virus-antibody mixture (Fig 2B). As a control, we used the same amount of a non-neutralizing antibody. Compared to the control antibody, the neutralizing antibody markedly inhibited vacuolization at 20 hpi (Fig 2C), indicating that virus binding to the cellular receptor is necessary for early-wave vacuolization. In the second neutralization experiment, we aimed to neutralize the released progeny virus and thereby prevent late-wave vacuolization without affecting the ongoing BKPyV replication. In short, we infected RPTECs with BKPyV (MOI 1), added the neutralizing antibody at 24 and 72 hpi, and imaged the cells at 96 hpi using widefield microscopy (Fig 2D). Here, we used a MOI of 1, which was not enough virus to induce early-wave vacuolization but yielded subsequent late-wave vacuolization. At the time of the first antibody addition, the early BKPyV protein large tumor antigen (LTag) is expressed, while progeny release has not yet occurred [27]. As expected, the neutralizing antibody markedly reduced late-wave vacuolization (Fig 2E). These findings indicate that progeny release and subsequent receptor binding are necessary for late-wave vacuolization. To further investigate the importance of receptor binding, we examined the effect of neuraminidase. This enzyme cleaves sialic acids on the cell surface and thus is known to inhibit BKPyV receptor binding [36,37]. As expected, pre-treatment with neuraminidase before BKPyV infection (MOI 200) markedly reduced early-wave vacuolization (Fig 2F). Next, we assessed if non-infectious BKPyV Vp1-virus-like-particles (BKVLPs) could induce early-wave vacuolization. When BKVLPs, in an amount equivalent to BKPyV MOI 200, were added to RPTECs, massive vacuolization was observed at 20 hpi (Fig 2G). Finally, we wanted to confirm the need for progeny release for the induction of late-wave vacuolization. When BKVLPs at ~MOI 200 were added, massive vacuolization was observed at 20 hpi, but this was

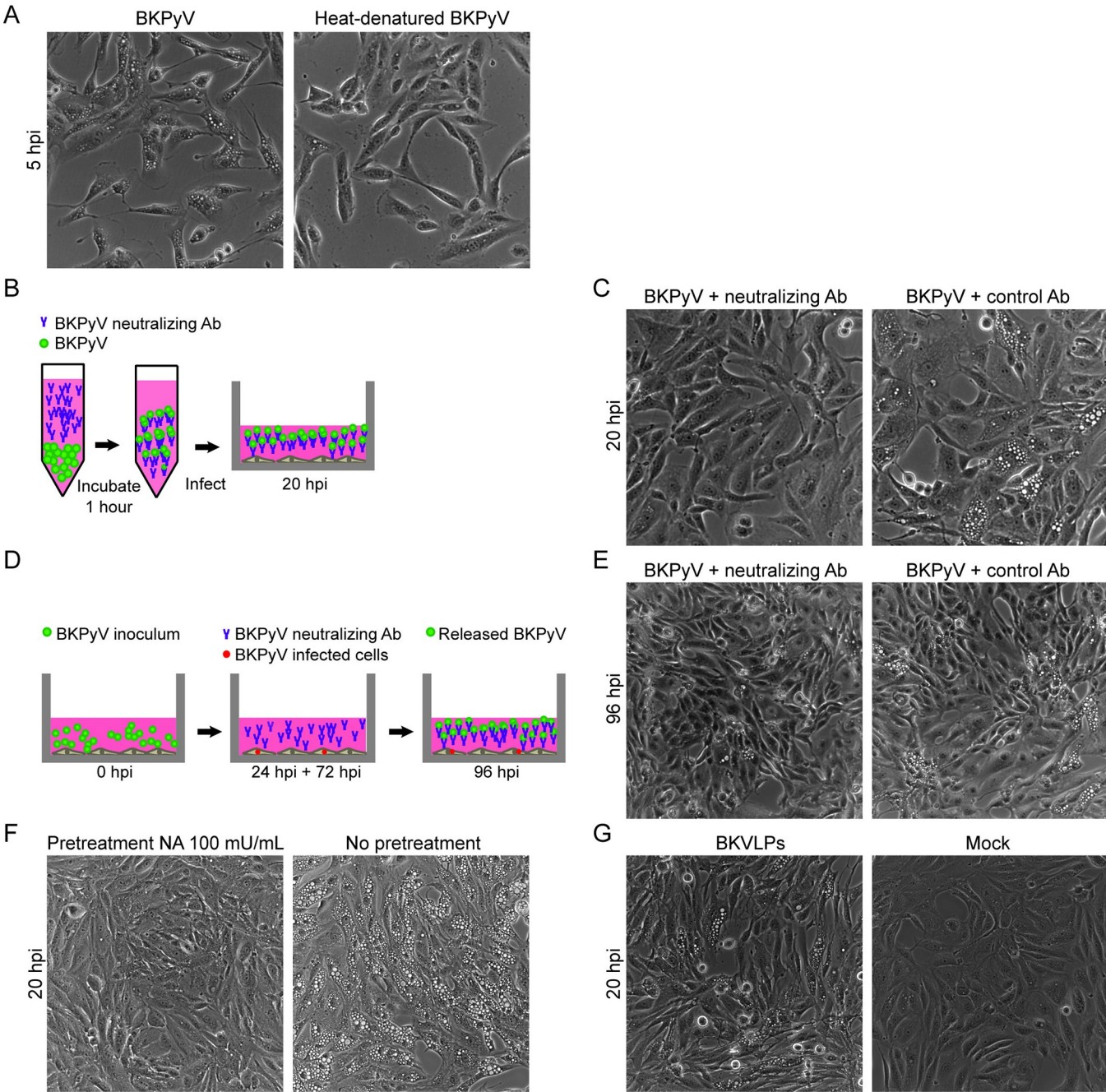

**Fig 2. Binding of intact viral particles or VLPs is necessary for induction of vacuolization.** (A) Phase-contrast images of RPTECs at 5 hpi after infection with untreated and heat-treated BKPyV (MOI 1000). Representative images from two independent experiments are shown. (B) Scheme illustrating the neutralizing experiment in panel C. The BKPyV inoculum (MOI 200) was preincubated with antibodies, and the mix was used to infect RPTECs. (C) Phase-contrast images of RPTECs at 20 hpi, using a BKPyV inoculum preincubated with a BKPyV-specific neutralizing antibody or a control antibody. Mock-infected RPTECs were included as a control. Representative images from three independent experiments are shown. (D) Scheme illustrating the neutralization experiment in panel E. RPTECs were infected with BKPyV (MOI 1), and neutralizing antibodies were added at 24 and 72 hpi. A control antibody was added to separate wells. (E) Phase-contrast images of RPTECs at 96 hpi, taken from wells with the addition of the BKPyV neutralizing antibody or the control antibody. Representative images from three independent experiments are shown. (F) Phase-contrast images of RPTECs at 20 hpi, taken from wells pretreated for two hours with 100 mU/mL neuraminidase before infection with BKPyV (MOI 200). A separate well without neuraminidase pretreatment was infected with BKPyV (MOI 200) and was used as a control. Representative images from three independent experiments are shown. (G) Phase-contrast images of RPTECs at 20 hpi, taken from wells infected with BKVLPs (equal to MOI 200 based on hemagglutination titer) or Mock-infected. Representative images from three independent experiments are shown. Illustrations in B and D were made by the authors.

transient, and no vacuolization was seen at 96 hpi (**S4 Fig, left panel**). However, when a combination of BKVLPs at ~MOI 200 and BKPyV at MOI 1 were added, vacuolization at both 20 hpi and 96 hpi was seen (**S4 Fig, middle panel**). Ultimately, when a combination of BKVLPs at ~MOI 1 and BKPyV at MOI 1 were added, no vacuolization at 20 hpi, but massive vacuolization at 96 hpi was seen (**S4 Fig, right panel**).

We conclude that BKPyV-induced vacuolization in RPTECs requires substantial viral binding and uptake and that BKPyV gene expression, replication and progeny release are only needed for late-wave vacuolization.

## Vacuoles represent enlarged endo-/lysosomes

To understand the mechanism behind vacuolization, we set out to determine which organelles the vacuoles were derived from by performing live-cell microscopy with different organelle markers. First, we infected RPTECs with an infectious dose known to induce early-wave vacuoles (MOI 100–200) and added Texas Red conjugated 70 kDa dextran, a molecule that is mainly internalized by macropinocytosis before it is routed through the endocytic pathway and finally ends up in endolysomes [38–40]. Endolysosomes are digestive lysosomes generated by a transient or complete fusion between late endosomes and lysosomes and, therefore, exhibit shared features, including GTPase Rab7, Lamp1, acidic pH, and hydrolytic enzymes such as cathepsin proteases [40–42]. Live-cell microscopy at 6 hpi demonstrated cells with numerous vacuoles, and the majority of vacuoles were dextran positive (**Fig 3A**), suggesting that vacuoles were mainly of endo-/lysosomal origin, even if they were enlarged compared to the dextran-positive structures in mock-infected RPTECs.

Next, we transfected RPTECs to transiently express fluorescently tagged endo-/lysosomal proteins. We used GTPase Rab5 as a marker for early endosomes and GTPase Rab7 and lysosomal-associated membrane protein 1 (Lamp1) as markers for late endosomes, endolysosomes, and lysosomes, respectively [43–45]. It should be noted that although the markers are prototypical markers of early endosomes, late endosomes, and lysosomes, they are not exclusively located on these structures and show some overlap. Two to three days post-transfection, the RPTECs were infected (MOI 100–200), and live-cell microscopy was performed at 6 hpi. This demonstrated some Rab5-positive vacuoles (**Fig 3B**), but the majority of vacuoles were positive for Rab7 (**Fig 3C**) and Lamp1 (**Fig 3D**). The vacuoles were negative for a fluorescent luminal marker of the endoplasmic reticulum (ER), as the ER was distributed between the Rab7-positive vacuoles (**Fig 3E**). Live-cell confocal microscopy yielded similar results (**S5A, S5B and S5C Fig**). As an additional marker for lysosomes, we utilized Lysoview 488 and 633. These fluorescent dyes are protonated in the low pH environment of lysosomes, but according to the producer, they also stain the less acidic early and late endosomes. The vacuoles were positive for Lysoview 488 and 633 (**Fig 4A**). In contrast, the vacuoles were negative for the fluorescent mitochondrial stain Mitoview 633 (**Fig 4B**). Taken together, the dextran and Lysoview staining and the Rab5, Rab7, and Lamp1 expression strongly suggest that the early-wave vacuoles are of endo-/lysosomal origin.

Next, we investigated if the late-wave vacuoles were of endo-/lysosomal origin. We infected RPTECs with BKPyV at MOI 1 and pulsed them with Texas Red conjugated 70 kDa dextran, or infected RPTECs that transiently expressed fluorescently tagged Rab7, Lamp1, or ER. At 96 hpi, live-cell imaging revealed that late vacuoles were positive for dextran (**S6A Fig**), Rab7 (**S6B Fig**), and Lamp1 (**S6C Fig**) but not the ER-marker (**S6D Fig**), suggesting that also late-wave vacuoles were of endo-/lysosomal origin.

We conclude that vacuoles observed both early and late in infection exhibit the same traits and represent enlarged endo-/lysosomes.

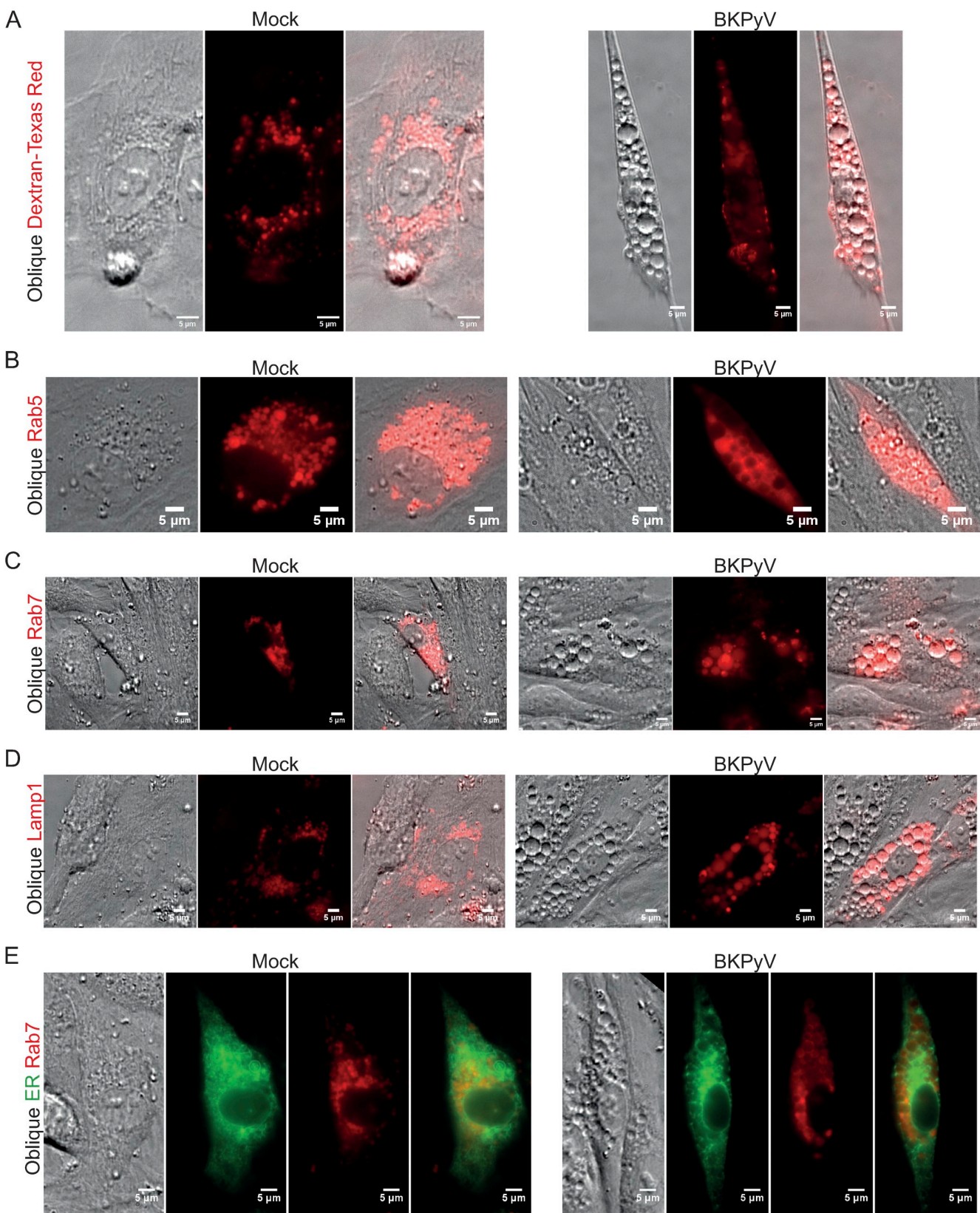

**Fig 3. BKPyV-induced vacuoles are of endo-/lysosomal origin.** Live-cell oblique contrast and fluorescence widefield microscopy of mock-infected and BKPyV-infected RPTECs (MOI 100–200) with markers of endo-/lysosomes and the ER at about 6 hpi. (A) RPTECs pulsed with Texas Red conjugated dextran (red) shortly after BKPyV infection. Representative images from four independent experiments are shown. (B) RPTECs that transiently express Rab5-mCherry, a marker of early endosomes. Representative images from two independent experiments are shown. (C) Transient expression of Rab7-mCherry (red), a marker of late endosomes and lysosomes. Representative images from four independent experiments are shown. (D) Transient expression of Lamp1-mCherry (red), a marker of lysosomes. Representative images from four independent experiments are shown. (E) Transient expression of mTurquoise2-ER (green) and Rab7-mCherry (red). Representative images from three independent experiments are shown. Scale bar = 5 µm for all images.

## The vacuoles contain viral particles, which suggests that BKPyV uptake is necessary for vacuolization

To examine if viral particles were directly associated with the vacuoles, we infected RPTECs (MOI 200) and performed immunofluorescence staining for the major capsid protein Vp1 and the early endosomal protein Early Endosome Antigen 1 (EEA1) at 1 hpi and 4 hpi, and Vp1 and the late endosomal and lysosomal protein Rab7 at 6 hpi. For the Vp1 staining, both a mouse monoclonal BKPyV Vp1 antibody and a polyclonal BKPyV Vp1 rabbit antiserum were used. While the monoclonal antibody preferably stains intact viral particles, the Vp1 antiserum also detects degraded particles (**S3B Fig**). Confocal microscopy at 1 hpi (**S7 Fig**) and 4 hpi (**Fig 5B and 5C**) demonstrated several vacuoles that were positive for EEA1, and independent of the Vp1 antibody used, Vp1 appeared to be mainly associated with the endosome membrane, even though in some endosomes, staining inside the lumen was also observed (**Fig 5B**). Using the monoclonal Vp1 antibody, the cytoplasm (**Fig 5B and 5C**) and the plasma membrane (**Fig 5C**) were also stained. Some vacuoles were only stained with the Vp1 antiserum, not the monoclonal Vp1 antibody or the EEA1 antibody (**Fig 5C**). This suggests that they were late endosomes or endolysosomes with partly degraded BKPyV. The vacuolized cells demonstrated the strongest Vp1 staining, which supports the notion that vacuolization is related to the binding and internalization of large amounts of BKPyV. At 6 hpi, we observed numerous Rab7-positive enlarged late endosomes, staining positive for Vp1 (**Fig 6A and 6B**). Again, the monoclonal Vp1 antibody showed additional staining of the plasma membrane and the cytoplasm (**Fig 6A**). Of note, whereas both EEA1 and Rab7 staining was circumferential and appeared to stain the entire endosome membrane, only parts of the membrane stained positive for Vp1 (**Figs 5B, 5C, 6A and 6B**). As expected, mock-infected cells only showed small early and late endosomes (**Figs 5A, 6A and 6B**).

To block the binding and uptake of BKPyV, we preincubated the infectious inoculum with neutralizing antibodies, infected the cells, and performed immunofluorescence staining at 6 hpi. In the control cells infected with an untreated BKPyV inoculum, the polyclonal Vp1 antiserum stained the membranes of the enlarged Rab7-positive late endosomes and the inside of some endosomes (**Fig 6C**). However, when antibody preincubation was performed, BKPyV uptake appeared strongly reduced, and vacuolization was lacking (**Fig 6D**).

Together, these results suggest that in some cells, the numerous viral particles that bind to the plasma membrane are internalized, enter the endocytic pathway, and induce a transient enlargement of endo-/lysosomes. As demonstrated by the differential staining of the Vp1 antiserum and monoclonal antibody, the viral particles appear to undergo some degradation or conformational change, but complete cargo degradation may be delayed.

To confirm that BKPyV was inside the vacuoles, we utilized transmission electron microscopy and examined the ultrastructure of vacuolized RPTECs (MOI 500) at 1 hpi and 4 hpi. At both timepoints, the plasma membrane was covered with viral particles (**Fig 7A and 7B**). Viral particles were also seen inside the cells, in monopinocytic vesicles and large vesicles containing multiple viral particles (**Fig 7B**). Overall, the cells had numerous vesicles limited by a single

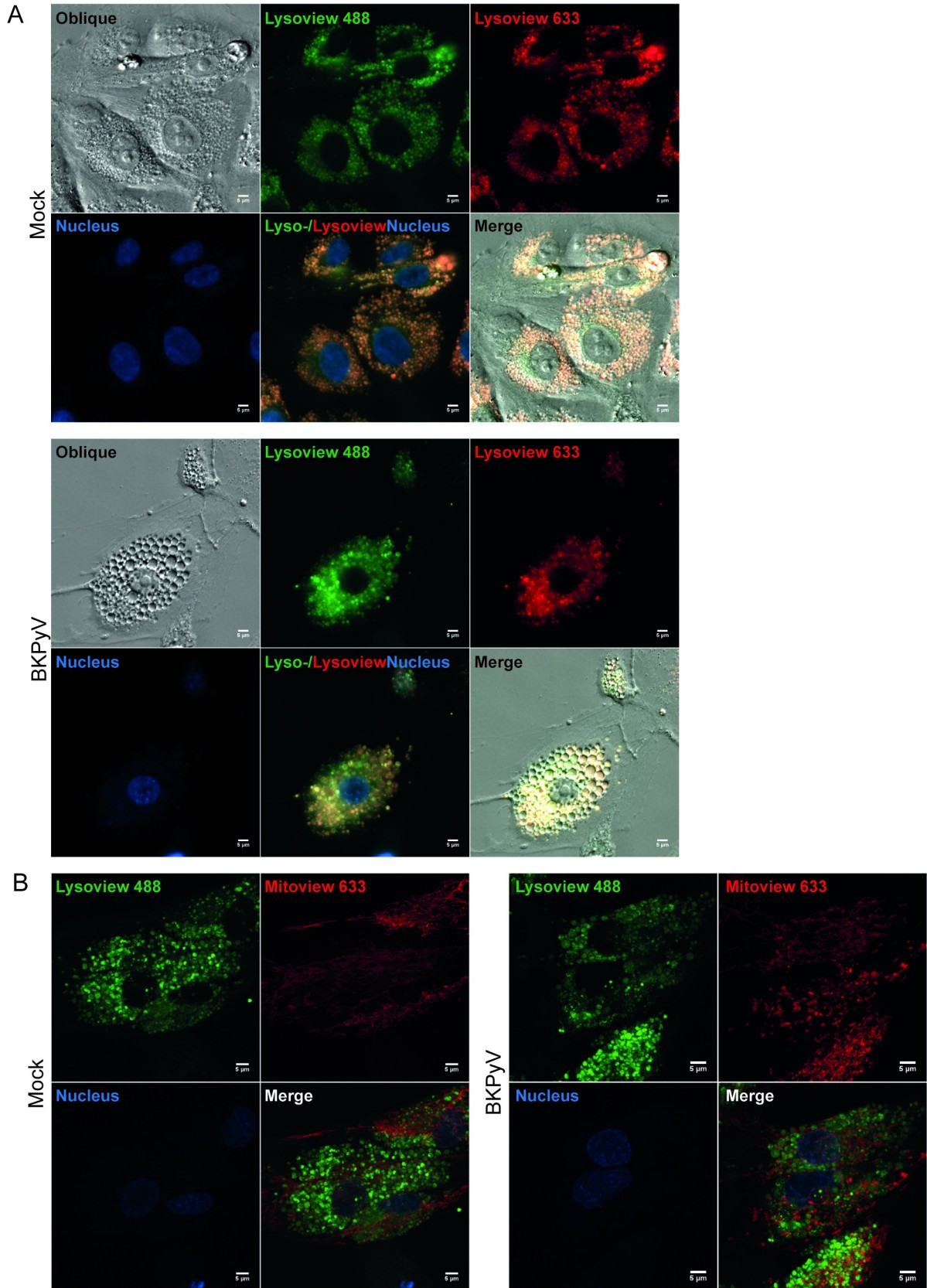

**Fig 4. BKPyV-induced vacuoles are positive for Lysoview and negative for Mitoview dye.** Live-cell widefield microscopy (A) and live-cell confocal microscopy (B) of mock-infected and BKPyV-infected RPTECs (MOI 200) at 6 hpi. The cells have been stained with A) Lysoview 488 (green) and 633 (red) dye, which labels endosomes and lysosomes. Representative images from three independent experiments are shown. B) Lysoview 488 (green) and Mitoview 633 (red), the latter dye labels mitochondria. Representative images from two independent experiments are shown. Nuclei are stained with Hoechst (blue) in all images. All scale bars = 5 μm.

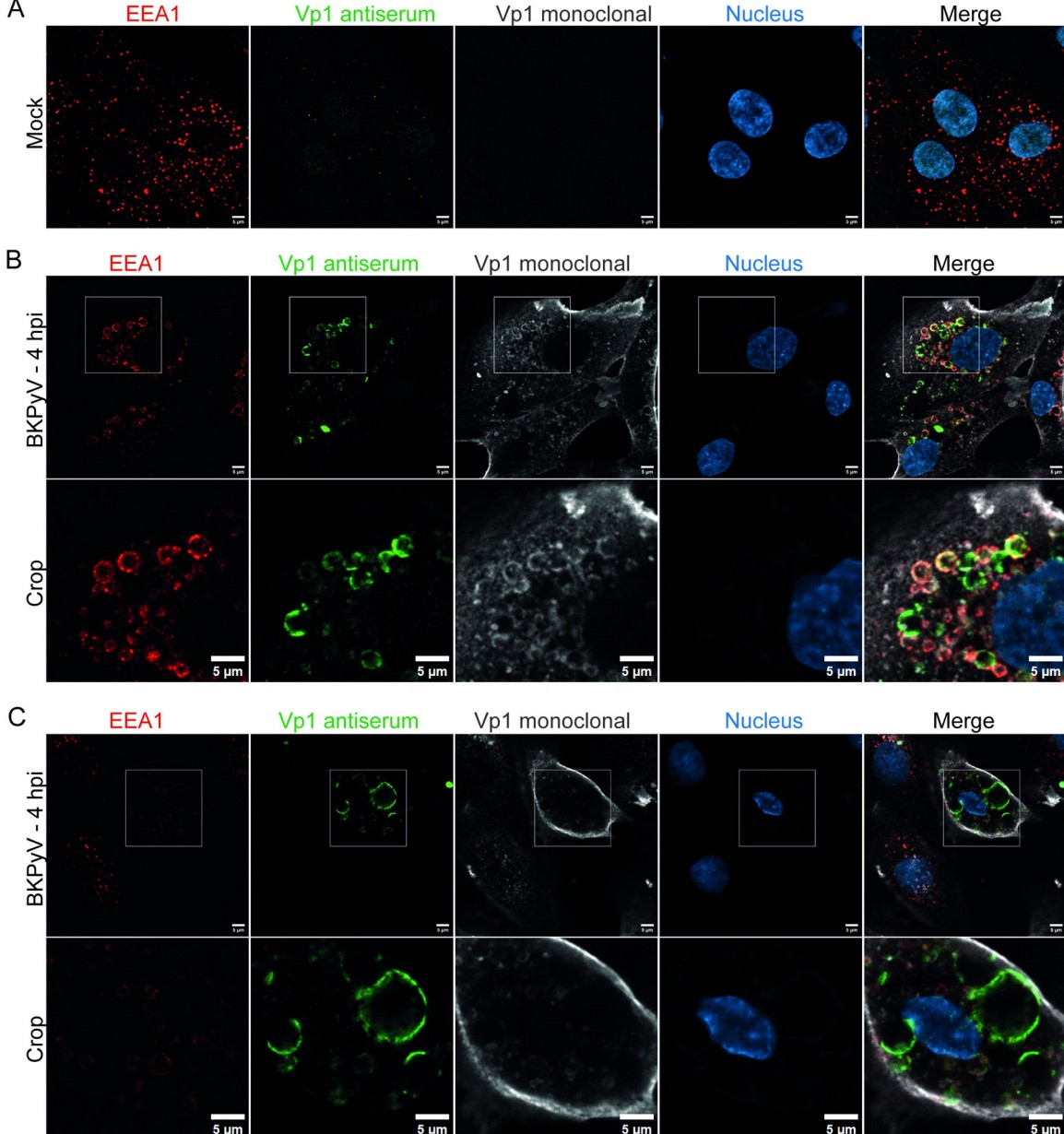

**Fig 5. BKPyV Vp1 is associated with early endosomes.** Confocal microscopy of mock-infected (A) and BKPyV-infected (MOI 200) (B) and (C) RPTECs at 4 hpi following immunofluorescence staining. A mouse monoclonal antibody against the early endosomal protein Early Endosome Antigen 1 (EEA1) (red) was used in combination with a rabbit BKPyV Vp1 antiserum (green) and a mouse monoclonal antibody against BKPyV Vp1 (grey). Nuclei are stained with DAPI (blue) in all images. Representative images from two experiments are shown. Scalebar = 5 μm for all images.

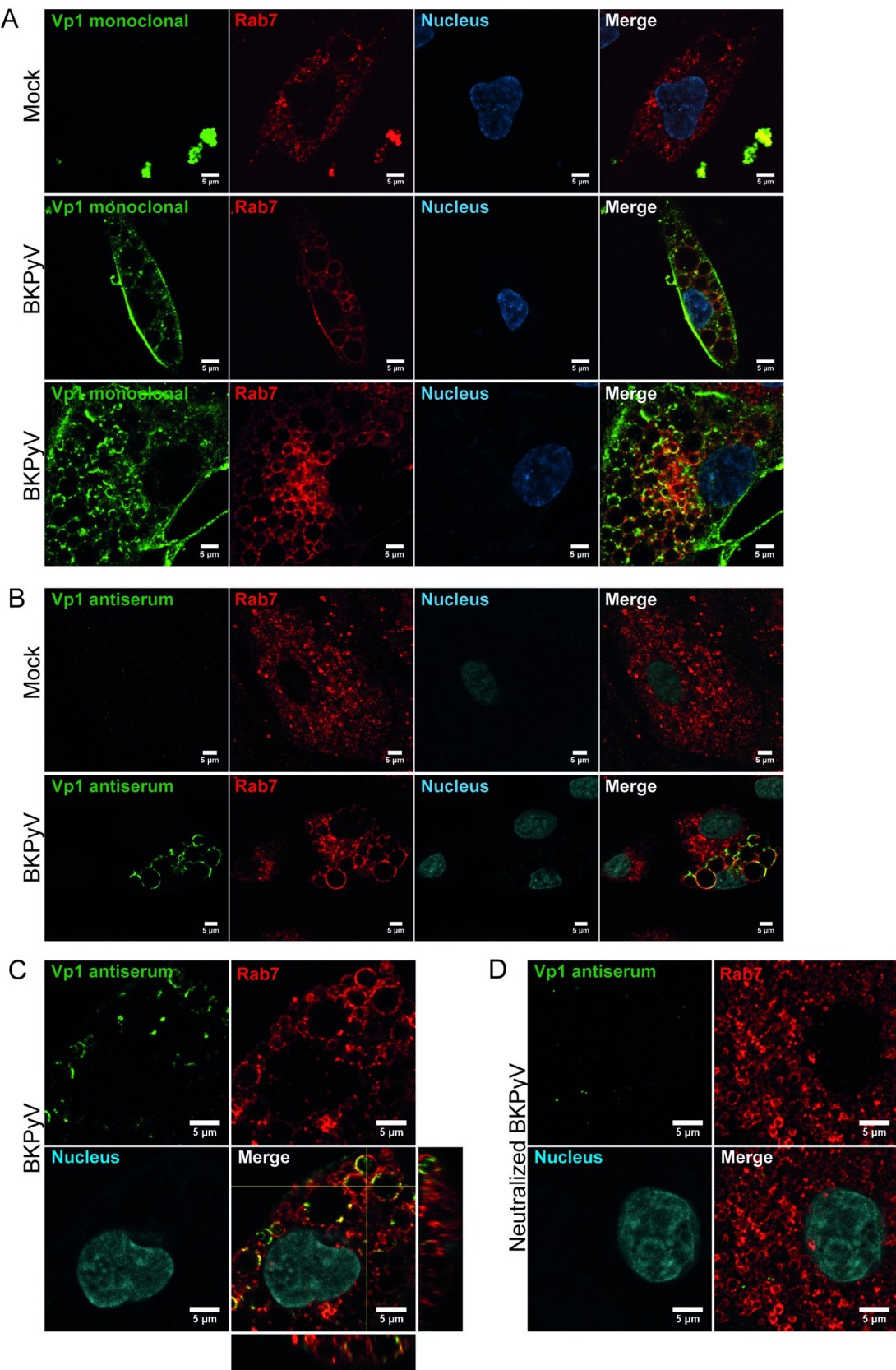

**Fig 6. BKPyV Vp1 is associated with enlarged endo-/lysosomes.** Confocal microscopy following immunofluorescence staining at 6 hpi of mock-infected and BKPyV-infected RPTECs (MOI 200) and RPTECs infected with BKPyV (MOI 200) preincubated with a BKPyV-specific neutralizing antibody. A mouse monoclonal antibody against Rab7 (red) was used in combination with: (A) A mouse monoclonal antibody against BKPyV Vp1 (green). Representative images from two independent experiments are shown. (B) A rabbit antiserum against BKPyV Vp1 (green). Representative images from three independent experiments are shown. (C and D) A rabbit antiserum against BKPyV Vp1 (green). Representative cells from a single experiment are shown. Nuclei are stained with Draq5 or DAPI (blue or cyan). Scalebar = 5 μm for all images.

membrane (**Fig 7C**). Most were translucent with little content except viral particles lining the inside of the membrane (**Fig 7D**) and were, therefore, similar to early endosomes [46]. We observed fusion events between smaller virus-containing early endosomes with larger virus-containing vacuoles (**Fig 7E**). Additionally, we observed viral particles in what looked more like late endosomes, endolysosomes, or lysosomes due to the electron-dense content (**Fig 7F**) [46]. Of note, the viral particles did not line the complete circumference of most vacuoles, explaining that only parts of the vacuole membrane stained positive for Vp1 (**Figs 5 and 6**).

To establish a productive infection, BKPyV must traffic from the endosomes to the ER for uncoating, successfully escape the ER, and enter the nucleus [34,47]. For SV40, the ER escape is initiated by the partly uncoated virions with exposed Vp2, which bind and redistribute the ER membrane protein B-cell receptor-associated protein 31 (Bap31) into distinct virus-induced foci [48,49]. To explore the ER trafficking and escape, we infected RPTECs with BKPyV MOI 10, 100, and 1000 and performed immunofluorescent staining for Bap31 at 2, 6, and 18 hpi. Similar to what has been described for SV40-infected cells, virus-induced Bap31-foci were present at 6 hpi and 18 hpi (**Fig 8A**). As the Bap31-foci were largest at 18 hpi, the number of cells with at least one Bap31-foci and the total cell number was quantified. With a MOI of 10, 56% of the cells contained Bap31-foci, and with a MOI of 100, this increased to 94% before it slightly declined to 75% for MOI 1000 (**Fig 8B**). The number of Bap31-foci per cell increased from 2.5 for MOI 10 to 5.6 for MOI 100 and was 4.2 for MOI 1000 (**Fig 8C**).

We conclude that endo-/lysosomal vacuoles are generated by the entry of massive amounts of BKPyV into the endocytic pathway, that degradation of cargo BKPyV seems to start early after endocytosis and that some viral particles manage to avoid this and enter the ER.

## Cell death and progeny release precede late-wave vacuolization

To examine the relationship between cell death and BKPyV-induced vacuolization, we infected RPTECs with different MOIs (0.1–200), added CellTox Green, a membrane-impermeable DNA stain that only stains dead and damaged cells, and imaged the cells with time-lapse microscopy. The experiment confirmed our initial observations of early- and late-wave vacuolization. In RPTECs infected with MOI 200, we observed cytoplasmic vacuolization already from about 3 hpi (**Fig 9A** and **S1 Video**). Most early vacuoles disappeared before 36 hpi. Cell damage and death started to occur at around 40 hpi, and notably, late-wave vacuolization emerged after this (**Fig 9A** and **S1 Video**). Again, the highest infectious dose caused more extensive vacuolization than the lower infectious doses (**Fig 9B** and **S2–S4 Videos**). Using MOI 10, cell damage and death started at about 40 hpi, and vacuolization emerged a few hours later and was widespread by 55 hpi (**S2 Video**). Using MOI 1 and MOI 0.1, cell damage and death mainly emerged from around 50 hpi and 56 hpi, respectively (**S3** and **S4 Videos**), but in some fields of view, green cells emerged before 48 hpi. Interestingly, we noted that late-wave vacuolization typically occurred in cells surrounding damaged or dead infected cells only a few hours after lysis (**Figs 9, S8A** and **S1–S4 Videos**). This suggests that a high local progeny virus concentration around lysed cells causes late-wave vacuolization. Like early-wave vacuolization, late-wave vacuolization seems transient (**S8B Fig**), although some vacuolized cells died before

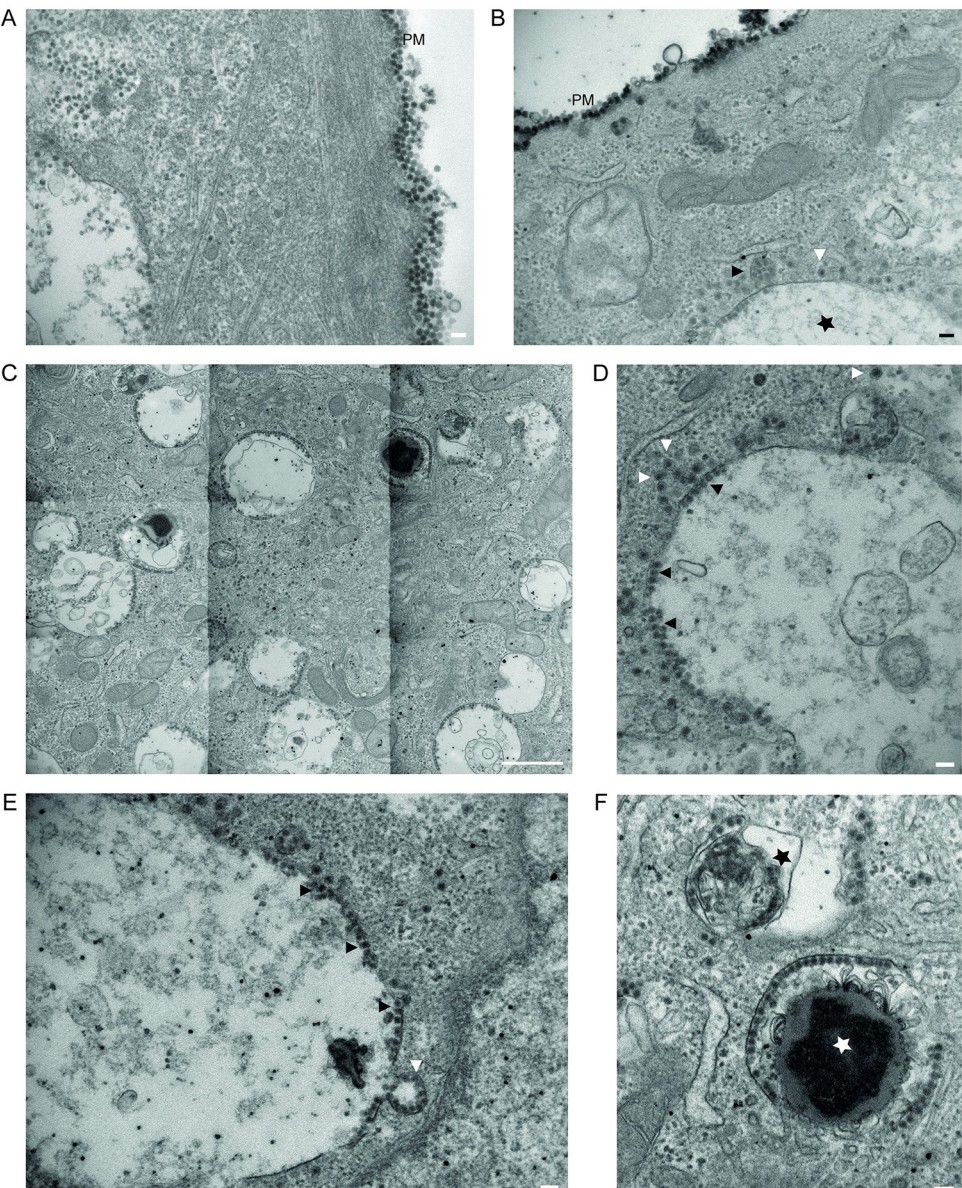

**Fig 7. Vacuoles contain BKPyV.** Transmission electron microscopy of BKPyV-infected RPTECs (MOI 500) at 1 hpi and 4 hpi. Viral particles on the plasma membrane (PM) (A) at 1 hpi. Scalebar = 100 nm. (B) Viral particles on the plasma membrane at 4 hpi. Intracellularly, monopinocytic vesicles with BKPyV (white arrowhead) and larger endosomes with several viral particles (black arrowhead). At the bottom of the image, parts of a single-membraned vacuole containing viral particles can be seen (black star). Scalebar = 100 nm. (C) Overview image demonstrating numerous translucent structures. Scale bar = 1 μm. (D) Translucent vacuole with little content. Note virus-containing monopinocytic vesicles close to the vacuole (white arrowheads) and virus inside the vacuole (black arrowheads). Scale bar = 100 nm. (E) Fusion event of endosome with numerous viral particles into an enlarged vacuole (white arrowhead). Note numerous membrane-attached viral particles inside the vacuole (black arrowheads). Scalebar = 100 nm. (F) Single-membraned structures with membranous content (black star) and electron-dense content (white star). Scale bar = 100 nm. All images are derived from a single experiment.

vacuolization was reversed. Mock-infected RPTECs did not demonstrate significant vacuolization (**S5 Video**). Moreover, adding the BKPyV-specific neutralizing antibody to BKPyV-infected RPTECs (MOI 0.1) at 24 hpi greatly reduced late-wave vacuolization (**S6 Video**).

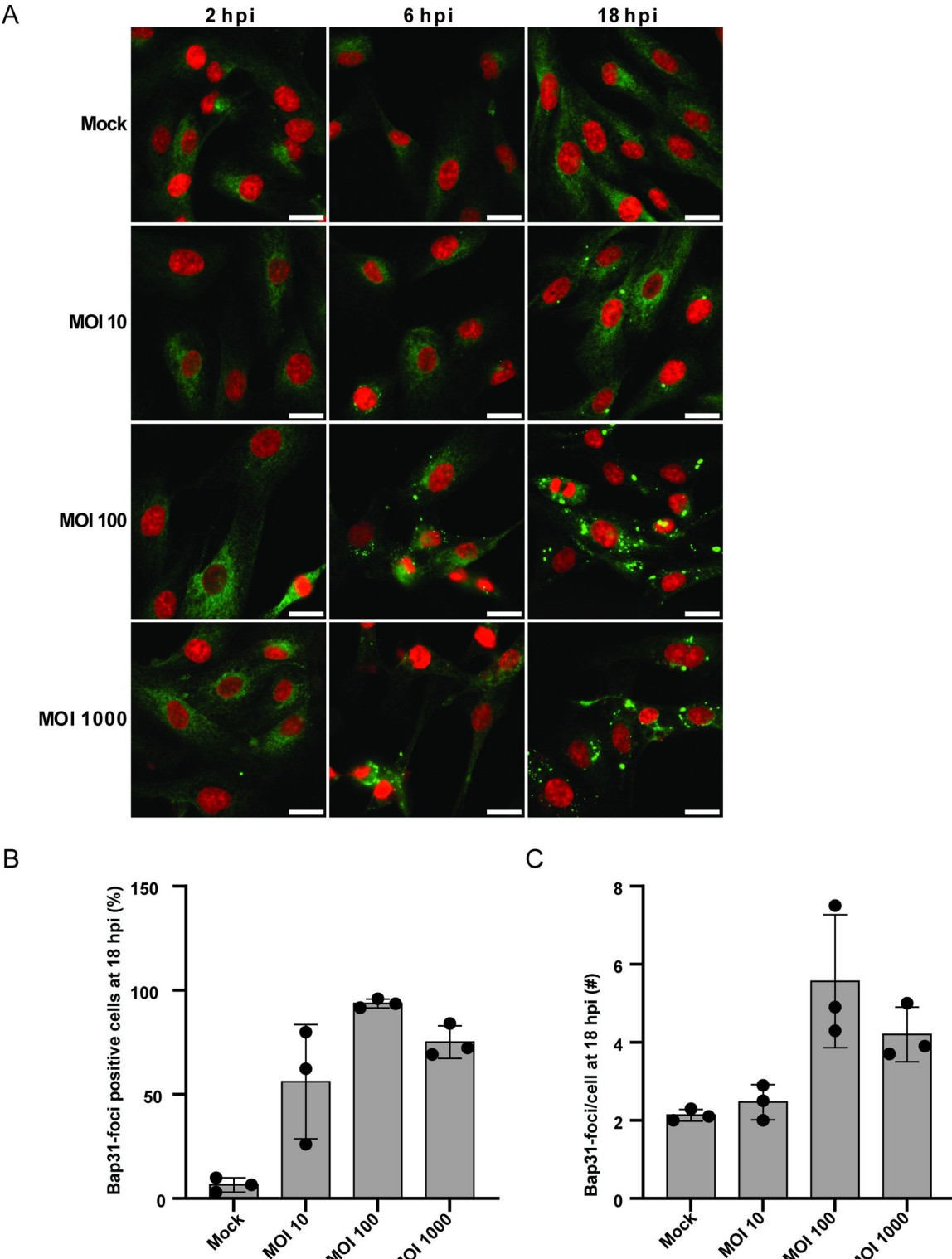

**Fig 8. BKPyV traffics to the ER and induces the formation of Bap31-foci.** (A) Immunofluorescence staining at indicated hpi and MOI of mock-infected and BKPyV-infected RPTECs. A mouse monoclonal antibody against Bap31 (green) was combined with DAPI staining of the nuclei (red). Representative images from three independent experiments are shown. (B) Mean percentage of Bap31-positive cells and (C) mean number of Bap31-foci per Bap31-positive cell in mock-infected and BKPyV-infected RPTECs (MOI 10, 100, and 1000) at 18 hpi. Error bars represent SD, and the value of each biological replicate (n = 3) is marked with circles. Scalebar = 20 μm for all images.

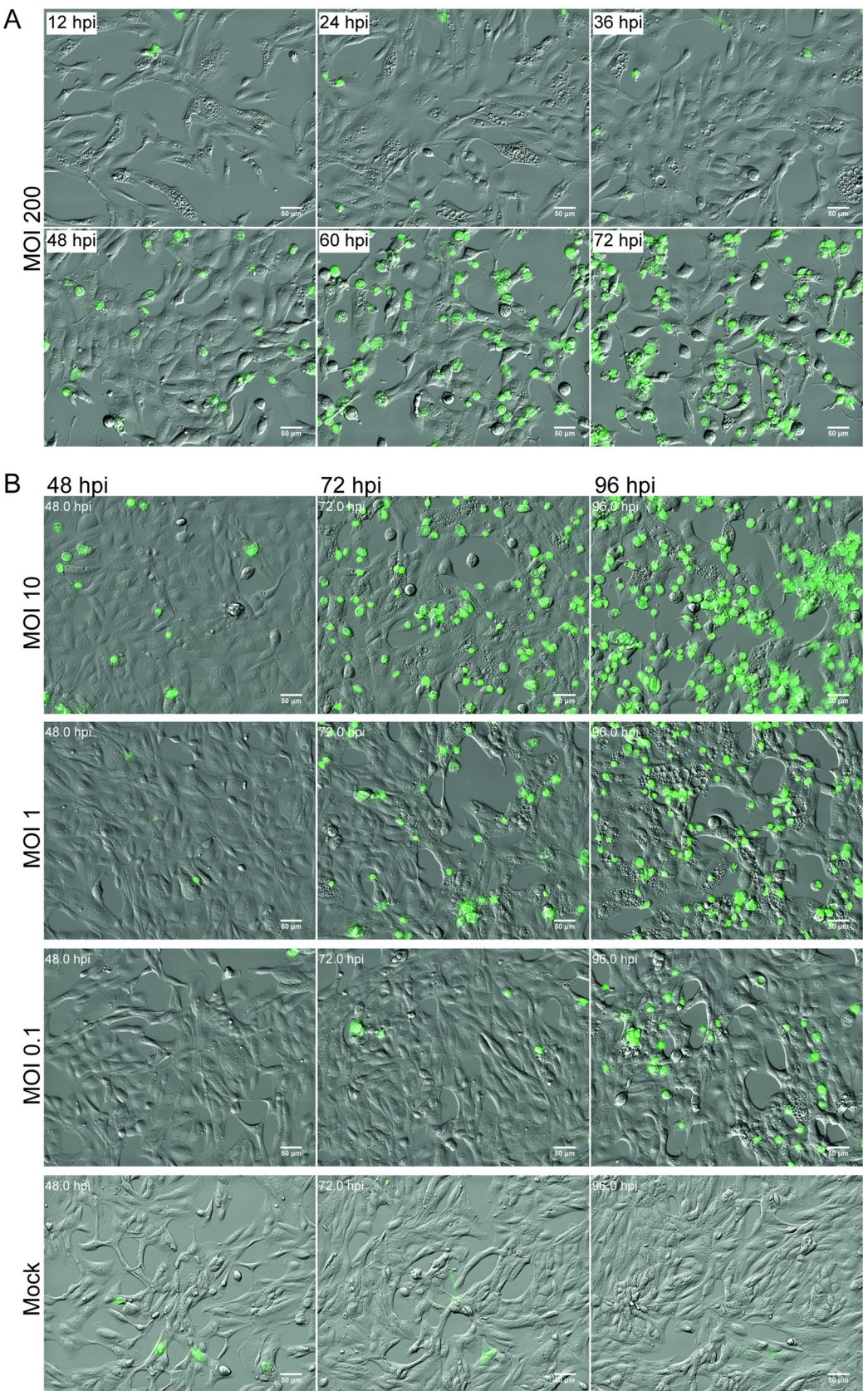

**Fig 9. Cell death precedes BKPyV-induced late vacuolization.** Selected merged images from time-lapse microscopy with oblique contrast and CellTox staining (green) of mock-infected and BKPyV-infected RPTECs. (A) MOI 200. (B) MOI 10, 1, 0.1, and mock. Scale bar = 50 μm.

To investigate the release of progeny virus, we infected RPTECs with BKPyV (MOI 0.1, 1, and 10) and measured the extracellular BKPyV DNA load in supernatants using a quantitative PCR. Independent of the inoculum used, at 48 hpi, we observed an approximately 2-log increase from the 24-hour input level. This suggests that progeny release started before 48 hpi (**Fig 10A**) and thus before vacuolization, which aligns with our time-lapse microscopy results. At 72 hpi, the BKPyV DNA load was further increased. Together, these results support that lytic cell death and progeny release precede vacuolization and that the exact timing depends on the infectious dose.

Lastly, we examined the cells that underwent late-wave vacuolization to determine if they were very recently infected and did not express BKPyV proteins, recently infected and only expressed BKPyV early proteins, or in a later stage of the replication cycle, expressing both early and late BKPyV proteins. We infected RPTECs with BKPyV (MOI 1) and performed immunofluorescence staining for LTag and Vp1 at 72 hpi. Widefield microscopy revealed that 57% of the vacuolized cells expressed LTag, while only 29% showed Vp1 expression, demonstrated as nuclear Vp1 staining (**Fig 10B**). Of note, some of the vacuolized cells demonstrated strong cytoplasmic Vp1 staining but lacked nuclear LTag and Vp1 staining (**S8C Fig**), suggesting that they had recently taken up BKPyV but that viral genes were not yet expressed.

We conclude that late-wave vacuolization occurs in very recently and recently infected cells surrounding lysed infected cells and, therefore, represents the same process as early-wave vacuolization. The focal pattern suggests that the bulk of BKPyV progeny is locally spread.

## Disruption of vacuolization has little impact on cell death and progeny release

SV40-induced vacuolization has been proposed to support lytic release via a positive feedback loop [18]. To examine if BKPyV-induced vacuolization could increase lytic release, we again utilized the BKPyV-specific neutralizing antibody to disrupt late vacuolization. First, we infected RPTECs (MOI 1) and added the BKPyV neutralizing antibody or a control antibody at 24 hpi and 48 hpi. Then, at 72 hpi, we measured released progeny by BKPyV quantitative PCR and counted CellTox-positive cells. The timing was vital since we aimed to measure the effect of disrupted late-wave vacuolization, not the effect of inhibiting new infections. Based on previous results (**Figs 9B and 10A, and S3 Video**), we expected the first progeny release to occur just before 48 hpi and, therefore, that none or few of the cells in the control wells that were infected by the progeny virus had time to complete the replication cycle and cause cell lysis before 72 hpi. As expected, the BKPyV-specific neutralizing antibodies, but not the control antibody, markedly reduced vacuolization (**S9 Fig**). However, we detected no significant differences in progeny release and cell death between the wells, i.e., without or with prominent late-wave vacuolization (**Fig 10C and 10D**). This result agrees with the observations made during the CellTox time-lapse experiments.

We conclude that vacuolization does not increase host cell lysis or release of BKPyV progeny.

## Vacuolization is blocked by inhibition of the vacuolar H$^+$ ATPase

To better understand the mechanism behind vacuolization, we treated RPTECs with four inhibitors known to affect cellular functions that might be necessary for vacuolization. In

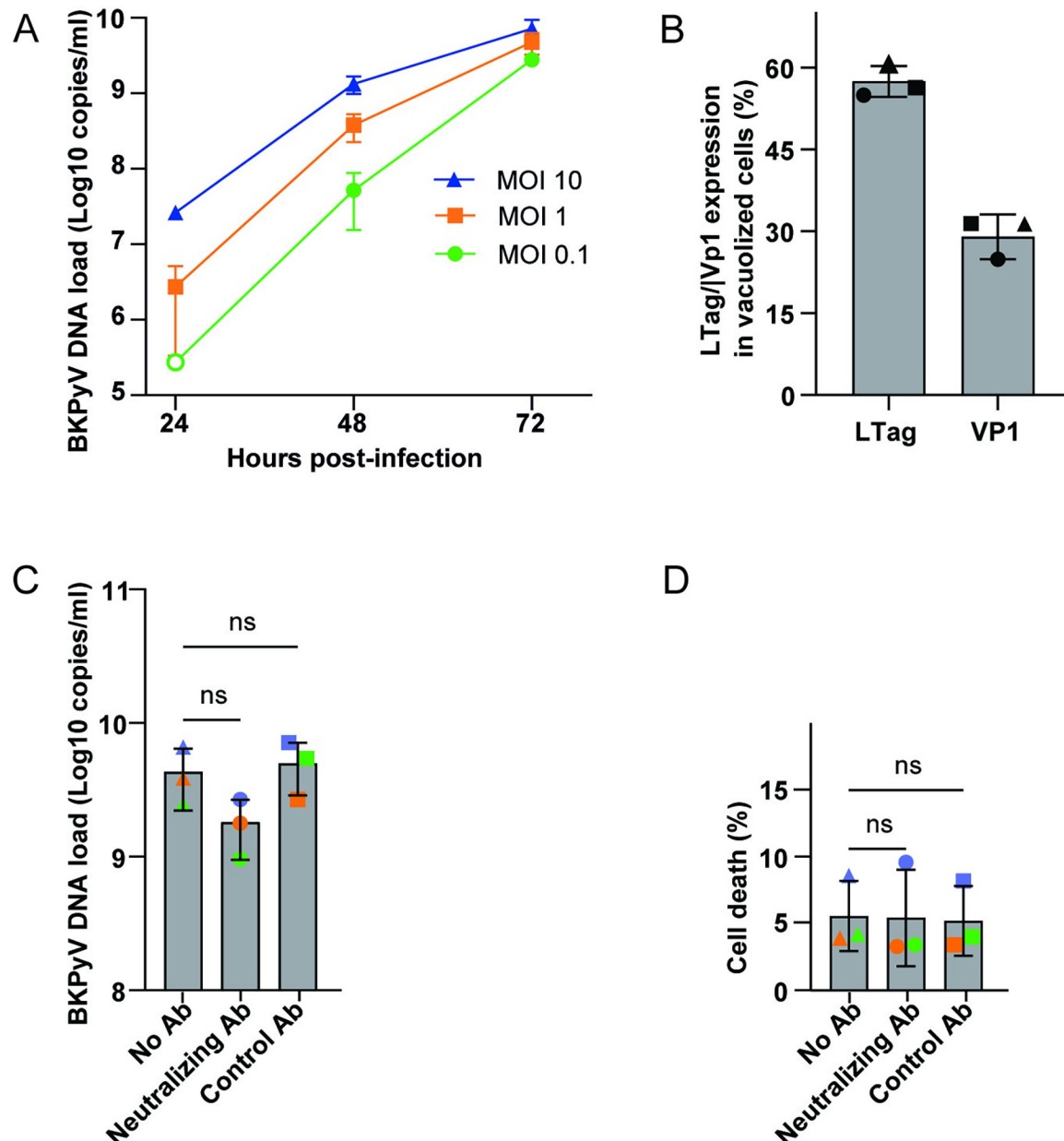

**Fig 10. Cell death and progeny release precede cytoplasmic vacuolization.** (A) RPTECs were infected (MOI 0.1, 1, and 10), and supernatants were collected at 24, 48, and 72 hpi for measurement of the extracellular BKPyV DNA load (log10 copies/ml) by quantitative PCR. The 24 hpi time point was used as input. As the input for MOI 0.1 was below the detection limit, a hypothetical value 10 times lower than the input of MOI 1 was plotted (open circle). Error bars represent SD and n = 3. (B) RPTECs were infected (MOI 1), and at 72 hpi, immunofluorescence staining was performed to determine if the vacuolized cells expressed nuclear LTag and Vp1. In total, 1393 vacuolized cells from three biological replicates, shown by triangles, circles, or squares, respectively, were analyzed. Error bars represent SD. (C) and (D) RPTECs were infected (MOI 1), and from 24 hpi, a BKPyV-specific neutralizing antibody, control antibody, or no antibody was added. At 72 hpi, supernatants were harvested, and the extracellular BKPyV DNA load (log10 copies/ml) was measured by quantitative PCR (C), and cell death (%) was determined by fluorescence microscopy with CellTox and Hoechst staining (D). Error bars represent SD, and each biological replicate (n = 3) is marked with different colors. ns = p-value > 0.05 by two-way ANOVA.

short, RPTECs were infected (MOI 200), and at 2 hpi, the inhibitor or the solvent DMSO was added. At 6 hpi, vacuolization of live cells was assessed by widefield microscopy.

Both the Rac1-inhibitor EHT1864 and the c-Jun N-terminal kinase (JNK) inhibitor Sp600125 have previously been reported to inhibit SV40 vacuolization [18]. Compared to DMSO, EHT1864 at 25 μM caused a 32% decrease in vacuolized cells, while Sp600125 at 10 μM showed no inhibitory effect (**Fig 11A**). In addition, we investigated the impact on BKPyV replication. At 48 hpi, immunofluorescence staining for LTag and agnoprotein was performed, and cells with LTag staining were quantitated. Treatment with EHT1864 from 2 hpi reduced the number of LTag-expressing cells by 35%, while treatment with Sp600125 reduced the number of LTag-expressing cells by 50% (**S10A and S10B Fig**). Moreover, both drugs strongly inhibited late gene expression as only a few cells expressed agnoprotein (**S10A and S10B Fig**). Sp600125 was difficult to dissolve, showed some cytotoxic effects, and was therefore not tested further. Of note, the addition of EHT1864 at 24 hpi inhibited both expression of Vp1 and agnoprotein at 48 hpi (**S10C Fig**) and almost completely blocked late-wave vacuolization (**S10D Fig**).

Methuosis is a cell death where the cells exhibit massive cytoplasmic vacuoles due to excessive macropinocytosis [50]. The sodium-hydrogen exchanger 3 inhibitor ethylisopropyla-miloride (EIPA) inhibits methuosis, macropinocytosis [51], and also the viral entry of viruses that exploit macropinocytosis, such as Vaccinia virus [52]. Furthermore, human papillomavirus and adeno-associated virus 2 have been reported to be sensitive to EIPA treatment as it perturbs virus entry via a ligand-induced endocytic pathway related to macropinocytosis [53] and the CLIC/GEEC pathway [54], respectively. Treatment with EIPA at 10 μM from 2 hpi increased the number of vacuolized cells by 22% (**Fig 11A**) and had no apparent inhibitory effect on BKPyV replication (**S10E Fig**). To rule out that the limited influence on vacuolization was caused by too late drug addition, we next performed a 24-hour pretreatment, followed by a 24-hour treatment with EIPA and EHT1864, respectively. At 24 hpi, widefield microscopy demonstrated a similar effect as when treatment started from 2 hpi (**S10F Fig**).

Bafilomycin A1 is an inhibitor of the vacuolar H$^+$ ATPase (V-ATPase) [55] and is known to disrupt endosomal acidification and block endo-/lysosomal maturation in the endocytic pathway [56–59]. Bafilomycin A1 also inhibits autophagic flux [60] and cytoplasmic vacuolization due to both methuosis [61, 62] and PIK-FYVE inhibition [63, 64]. Strikingly, treatment with bafilomycin A1 at 20 or 200 nM from 2 hpi almost completely blocked vacuolization (**Fig 11A and 11B**), apparently without affecting BKPyV replication (**S11A and S11D Fig**). However, bafilomycin A1 treatment from 0 to 2 hpi blocked vacuolization and inhibited BKPyV replication strongly (**Figs 11B and S11B**). Extending the treatment up to 48 hpi, the inhibitory effect on BKPyV replication was even stronger (**S11C Fig**). Transmission electron microscopy at 4 hours following infection and treatment with bafilomycin A1 at 200 nM, revealed that BKPyV was taken up by the cells but that the viral particles were mainly present in monopinocytic vesicles (**Fig 11C**) and not in enlarged endo-/lysosomes, as observed for untreated cells (**Fig 7**).

We conclude that an active V-ATPase is necessary for BKPyV-induced vacuolization as bafilomycin A1 treatment almost completely prevented early-wave vacuolization. In addition, it inhibited one or more early steps in the BKPyV replication cycle. EHT1864 only caused a modest reduction in early-wave vacuolization but appeared to inhibit BKPyV-induced late-wave vacuolization and BKPyV replication.

## Discussion

Although cytoplasmic vacuolization is a well-known cytopathic effect of SV40 replication in monkey kidney cells [16,17], it was unclear if the closely related human polyomavirus BKPyV

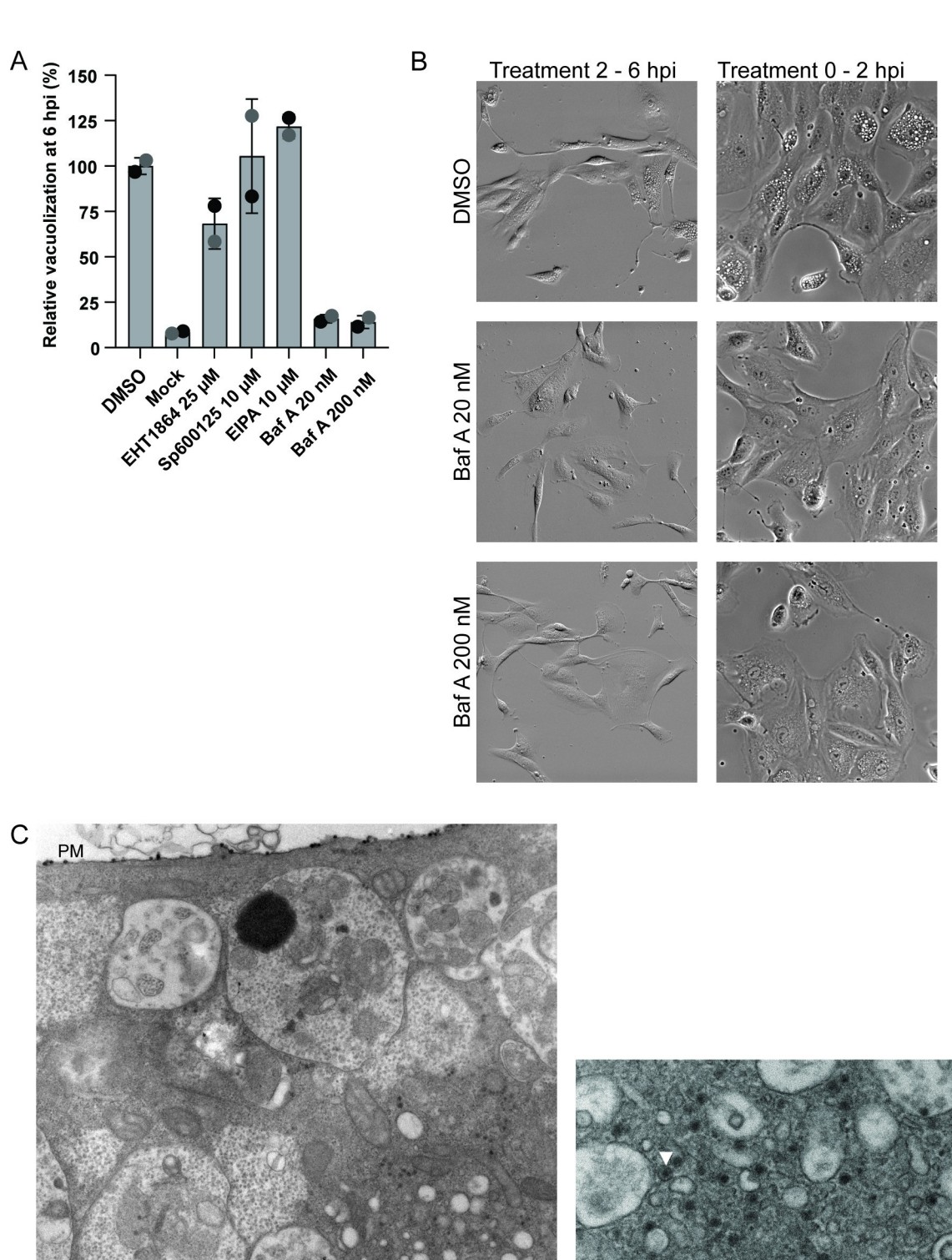

**Fig 11. Bafilomycin A1 inhibits BKPyV-induced vacuolization.** (A) Quantification of vacuolization in BKPyV-infected RPTECs (MOI 200) at 6 hpi following treatment with EHT1864 at 25 μm, SP600125 at 10 μm, EIPA at 10 μm, bafilomycin A1 (Baf A) at 20 and 200 μm, and DMSO. Treatment was started at 2 hpi. Data represents relative vacuolization. Vacuolization was examined by widefield microscopy and normalized to the DMSO control. Error bars represent SD, n = 2. At least 300 cells have been counted per treatment. Representative widefield images are shown in (B) left: treatment from 2–6 hpi, and imaged at 6 hpi, right: treatment from 0–2 hpi, and imaged at 5 hpi. Representative images from two independent experiments are shown (C) Transmission electron microscopy of BKPyV-infected RPTECs (MOI 500) at 4 hpi following Baf A 200 nM treatment from 0 hpi. Note virus-containing monopinocytic vesicles (white arrowheads). Scale bar = 100 nm. Images are derived from a single experiment.

induces cytoplasmic vacuolization. In this study, we report that BKPyV induces cytoplasmic vacuolization in renal tubular epithelial cells, the host cells supporting BKPyV replication *in vivo* and *in vitro* [24]. The cytoplasmic vacuoles emerge in a subset of RPTECs as early as 3 hpi and disappear between 36–48 hpi, apparently without harming the cells. Vacuolization is only observed in cells with a massive entry of BKPyV, and the vacuoles are enlarged endo-/lysosomal structures that contain membrane-bound BKPyV. Independent of the size of the viral inoculum, when lytic cell death and progeny release occur, the next round of vacuolization starts, and this first happens in cells directly surrounding lysed cells. Notably, early- and late-wave vacuolization are identical processes induced by binding and uptake of BKPyV from the initial inoculum or released progeny virus, respectively, and consequently occur in cells that have been recently infected. In line with this, BKPyV-induced vacuolization does not increase lytic BKPyV release. Pre-treatment of the cells with neuraminidase prevents early-wave vacuolization, and preincubation of the viral inoculum with a BKPyV-specific neutralizing antibody or treatment of the infected cells with the same neutralizing antibody inhibits early- and late-wave vacuolization, respectively. Moreover, Bafilomycin A1 treatment concurrent with BKPyV addition inhibits vacuolization and replication. In contrast, bafilomycin A1 treatment starting from 2 hpi only inhibits vacuolization, suggesting that the formation of vacuoles depends on the continued activity of the vacuolar H+ V-ATPase. Interestingly, inhibitors successfully used to inhibit SV40-induced vacuolization had only limited effect. Overall, we show that a massive entry of BKPyV or BKVLPs into RPTECs induces transient cytoplasmic vacuolization. Late-wave vacuolization preferentially occurs in cells directly adjacent to lysed cells, suggesting that most released progeny spreads locally.

How do our results align with reports on SV40-induced vacuolization? Similar to what has been found for SV40, BKPyV-induced vacuolization depends on the binding of viral particles to the plasma membrane [16,17]. However, as BKPyV is known to bind to GD1b and GT1b and not to GM1, the vacuolization does probably not depend on the ganglioside GM1, as suggested for SV40 [16,24,65,66]. Like SV40-induced vacuolization, BKPyV-induced vacuolization is cell-type dependent as large amounts of BKPyV (MOI 100 and 200) did not induce vacuolization in HeLa and CV-1 cells, even if the Vp1-staining at 6 hpi suggested that BKPyV was internalized. The lack of vacuolization is puzzling but may be explained by a combination of lower levels of the attachment receptors, the gangliosides GD1b and GT1b [37], and different cell-type specific trafficking. In support of this, Bennett and colleagues [34] have shown that preincubation of CV-1 cells with GT1b increases the infectivity of BKPyV. Furthermore, Luo and colleagues [16] demonstrated that a GM1-binding BKPyV pseudovirus could induce vacuolization in CV-1 cells. Moreover, HeLa cells are known to have higher levels of ganglioside GM3 than GD1b and GT1b [67], and finally, differences in BKPyV trafficking in RPTECs compared to CV-1 and Vero cells have been described [34,68]. The BKPyV-induced early-wave vacuoles are transient and disappear before 48 hpi. Transient vacuolization has seemingly not been reported for SV40-infected CV-1 cells but for SV40-infected primary African green monkey kidney (AGMK) cells [17]. In addition, it has been reported for CV-1 cells after the addition of SV40 Vp1 pentamers [16]. Importantly, both BKPyV- and SV40-induced vacuoles

are endo-/lysosomal structures [18]. However, whereas we find that the vacuoles contain numerous membrane-bound viral particles, this has not been detected in SV40-infected CV-1 cells [16]. Neither was it detected in BKPyV-infected human embryonic fibroblasts [20]. One possible reason for this discrepancy is the later time points they investigated, 48 and 12 hpi, respectively, since our immunofluorescent staining indicates that BKPyV in vacuoles are partly degraded at 4 hpi. Further contrasting what was reported for SV40 [18], BKPyV-induced vacuolization did not increase cell death or progeny release. Moreover, the Rac1 inhibitor EHT1864 and the JNK inhibitor SP600125 only modestly inhibited or did not affect BKPyV-induced early-wave vacuolization, respectively, therefore do not support a role of the Rac1-JNK signaling pathway in vacuolization [18]. However, EHT1864 did inhibit late BKPyV-induced vacuolization, but this was probably an indirect effect due to the strong inhibitory effect on BKPyV replication, resulting in reduced progeny release. Further investigations of the antiviral effects of EHT 1864 on BKPyV replication are needed.

Although the vacuoles in both SV40-infected and BKPyV-infected kidney epithelial cells seem to arise from endo-/lysosomal structures, a recent kidney allograft biopsy study on patients with BKPyV nephropathy described cytoplasmic vacuolization that was mediated by ER stress [31]. ER stress typically occurs due to the accumulation of misfolded proteins and may include viral proteins [69]. Persistent ER stress and failure to restore ER homeostasis can lead to apoptosis [69] or paraptosis. The latter is a caspase-independent type of cell death where cells show extensive cytoplasmic vacuolization due to swelling of the ER [70]. Importantly, the cytoplasmic vacuoles formed during ER stress-induced paraptosis are of ER origin [70, 71]. In our study, the vacuoles were negative for a transiently expressed ER marker. Moreover, vacuoles appeared shortly after BKPyV entry and, therefore, before the translation of viral proteins. Lastly, our neutralization experiments demonstrated that neither the expression of viral proteins nor the progeny virus production was enough to induce late-wave vacuolization. Taken together, it is unlikely that ER stress played a role in BKPyV-induced vacuolization of RPTECs.

We propose that vacuolization is caused by a transient overload of the endocytic pathway due to the massive BKPyV entry in some cells. Uptake of BKPyV is known to occur via receptor-mediated endocytosis [72,73]. Based on our experiments with the macropinocytosis inhibitor EIPA, macropinocytosis is not involved. Similar results have been reported for SV40 and Merkel cell polyomavirus [53,74]. After internalization, endocytic vesicles with BKPyV seem to be targeted to early endosomes, as shown by colocalization of EEA1 and Vp1 and by transmission electron microscopy showing the presence of viral particles in structures resembling early endosomes. The early endosomes are clearly enlarged, possibly due to the observed fusions between early endosomes and other BKPyV-containing vesicles, resulting in vacuoles with the inside partly lined with BKPyV. Bafilomycin A1 treatment from 0 hpi appears to retain BKPyV in monopinocytic vesicles or early endosomes, and this inhibited both vacuolization and BKPyV replication, whereas treatment from 2 hpi only inhibited vacuolization. These results suggest that vacuole accumulation depends on a massive BKPyV uptake and continued endosome fusion. In untreated cells, the enlarged early endosomes matured to late endosomes, as shown by Vp1 and Rab7 colocalization, and the late endosomes matured into endolysosomes, as the vacuoles, in addition to Rab7 expression, expressed fluorescent Lamp1 and were stained by Lysoview. The differential Vp1 staining observed when different antibodies were used indicates that plasma-membrane-bound and cytoplasmatic BKPyV exhibit a different conformation than vacuole-associated BKPyV. This is possibly due to the starting degradation of BKPyV in the endolysosomal vacuoles. Degradation of the cargo could cause the reformation of lysosomes and contribute to the transient nature of the vacuoles [41].

Our results align with the current understanding of polyomavirus entry [18,74–78]. Bafilo-mycin A1 treatment from 0 to 24 hpi has been shown to inhibit BKPyV replication in Vero cells [79], and treatment from 0 hpi with ammonium chloride, an agent known to raise the pH in acidic compartments, blocked BKPyV entry in RPTECs [77]. In addition, bafilomycin A1 has been shown to disrupt the maturation of the cystic fibrosis transmembrane conductance regulator [80], an ion channel reported to be involved in early trafficking of BKPyV to the ER [81]. Bafilomycin A1 and ammonium chloride have also been shown to inhibit the replication of mouse polyomavirus, Merkel Cell polyomavirus, and SV40 [74–76,78]. In more detail, bafi-lomycin A1 is suggested to reduce SV40 internalization [75], inhibit the transition of SV40 and mouse polyomavirus from early to late endosomes [75,78], and disrupt a pH-dependent conformational change of the mouse polyomavirus capsid [76]. Importantly, for a successful, productive BKPyV infection, some viral particles must be transported to the ER [37,77]. Simi-larly, SV40 traffic through the endocytic pathway to the ER [75]. Additionally, mouse poly-omavirus has been suggested to traffic through the endocytic pathway to endolysosomes, where the capsids undergo a conformational change before ER delivery [76]. Our study sup-ports that BKPyV traffic through the endocytic pathway. This aligns well with a recent study where Rab18 was reported to facilitate the transport of BKPyV from late endosomes to the ER [47]. Based on the quantification of Bap31-foci, ER delivery seems to reach a plateau when a viral input between MOI 100 and 1000 is used, indicating that vacuolization does not facilitate ER delivery of BKPyV. We speculate that the majority of viral particles in endolysosomes do not traffic to the ER but are instead degraded and that viral particles that successfully initiate a productive infection do not necessarily follow the bulk of BKPyV to the endolysosomes but may instead be transported to the ER from early or late endosomes.

Although non-lytic release of BKPyV from host cells has been proposed [29,82], our results support lytic release as the main exit route since progeny release co-occurred with increasing cell death of infected cells. That late-wave vacuolization is initially focal and occurs in cells directly surrounding lysed cells may suggest a form of local cell-to-cell spread [83]. Conven-tional cell-to-cell spread typically occurs via cell-cell contacts and allows viral particles to be directly transmitted from an infected cell to an adjacent uninfected cell without being released into the extracellular environment. This way, the virus is typically inaccessible to antibodies and, therefore, resistant to antibody neutralization. In our experiments, the addition of a BKPyV-specific neutralizing antibody clearly reduced vacuolization, making conventional cell-to-cell spread less likely. Notably, late-wave vacuolization occurred even when a low MOI of 0.1 was initially used. This implies that cell lysis gives a very high local BKPyV concentration and suggests that the high MOIs used for early-wave vacuolization were physiologically relevant.

Possibly, retention of BKPyV on the surface of the lysed cell, as previously detected in heavily infected renal tubules [84] and on decoy-like cells [33], increases the local BKPyV con-centration and, thereby, the chance of infecting the neighboring cells. These viral particles would presumably be accessible to neutralizing antibodies, which would fit well with our observation of inhibited late-wave vacuolization. However, in immunocompromised kidney transplant patients, low concentrations or genotype mismatch of BKPyV-specific neutralizing antibodies [85] may hamper the neutralization of BKPyV. In line with this, focal BKPyV repli-cation is frequently observed in kidney allograft biopsies from patients with BKPyV nephropa-thy and may sometimes lead to false negative biopsies [86].

In conclusion, our study shows that infection of RPTECs with a massive dose of BKPyV induces transient endo-/lysosomal vacuolization shortly after viral entry. BKPyV-induced cytoplasmic vacuolization is, therefore, an early, although not necessary, event of the replica-tion cycle. We speculate that the vacuoles appear due to a transient viral overload of the

endocytic pathway. The appearance of vacuolized cells around lysed cells suggests a form of local virus spread that may also be important *in vivo*. Our study contributes new insight into the insufficiently understood replication cycle of BKPyV in RPTECs.

## Materials and methods

### Cells, virus, and infection

Primary human RPTECs from Lonza and Sciencell were cultured in renal epithelial growth medium (REGM; Lonza), while CV-1 and HeLa cells were cultured in Dulbecco's modified eagle medium (DMEM) with high glucose and 10% fetal bovine serum (FBS). To develop polarized RPTECs, RPTECs were cultured on cell culture inserts as recently described [33]. Cesium-chloride density gradient purified BKPyV of the Dunlop strain was used for all infections. BKPyV was purified from the supernatant of BKPyV-infected Vero cells using a protocol based on two previously published protocols [12,77]. Infections were performed for 2 hours at 37˚C, followed by removing the infectious inoculum and adding fresh REGM. The fluorescent focus units per ml were determined by infecting RPTECs with serial dilutions of the virus stock, followed by immunofluorescence staining and widefield microscopy, counting the cells expressing Vp1 at 72 hpi [87]. Polarized RPTECs were infected at 8 days post-seeding via the apical compartment for 2 hours, and this was followed by washing and the addition of fresh REGM. BKVLPs were generated at the University of Basel [88]. Infections with BKVLPs were performed identically as infections with purified BKPyV [89]. To examine early-wave vacuolization, high MOIs (MOI $\geq$ 100–200) had to be used for the induction of vacuolization in a sufficient number of cells. Furthermore, for low-throughput techniques such as transmission electron microscopy, an even higher MOI of 500 was used to facilitate imaging. To examine late-wave vacuolization a low MOI (MOI 0.1–10) was used, since it is the large amount of released progeny virus that causes the vacuolization. Finally, to study the impact of potential inhibitors on BKPyV replication, a MOI of 1 to 5 was used.

### Heat disassembly, antibody neutralization of viral particles, and neuraminidase treatment

To disassemble the viral particles, the BKPyV inoculum (MOI 1000) was heat-treated at 100˚C for five minutes before it was allowed to cool to 37˚C. An identical inoculum that had not been heated was used as a control. Infections were performed for 2 hours at 37˚C before the inoculum was replaced with fresh REGM. Vacuolization was assessed by widefield microscopy during the first 24 hpi.

Neutralization of the BKPyV inoculum was performed by preincubating the inoculum with a BKPyV-Vp1 specific neutralizing antibody (Virostat 4942) or a control antibody at the same concentration (~14 μg/ml) for one hour at 37˚C. After preincubation, RPTECs were infected with the inoculum-antibody mixture for 2 hours at 37˚C followed by removal of the inoculum and addition of fresh REGM. Vacuolization was then assessed by widefield microscopy during the first 24 hpi.

Inhibition of late-wave vacuolization was performed by first infecting RPTECs with BKPyV (MOI 1), and at 24 and 72 hpi, the BKPyV-Vp1 specific neutralizing antibody (Virostat 4942) or the control antibody was added to the cell medium. Vacuolization was then assessed by phase-contrast microscopy at 72 and 96 hpi. To further assess the effect of disrupted vacuolization, RPTECs were infected (MOI 1 and 5), then the same experiment was performed, except that the antibodies were added at 24 and 48 hpi. At 72 hpi, supernatants were first sampled for BKPyV qPCR before CellTox (1x) was added, and widefield microscopy was performed to determine the percentage of dead cells.

Neuraminidase treatment was performed by pre-treating RPTECs with 100 mU/mL neuraminidase from Clostridium perfringens, type V (Sigma-Aldrich) for two hours at 37°C prior to infection with purified BKPyV (MOI 200). At 20 hpi, RPTECs were imaged for vacuolization.

## Hemagglutination inhibition assay

The hemagglutination inhibition assay was adapted from Neel and colleagues [90]. In round-bottom microtiter wells, equal volumes of 4 hemagglutination units of purified BKPyV and serial dilutions of a BKPyV-Vp1 specific neutralizing antibody (Virostat 4942) were incubated for 1 hour at room temperature. After incubation, an equal volume of O-type erythrocytes was added, and the wells were incubated at 4°C for 1–2 hours. A positive control without antibodies and a negative control without virus was included. Sedimentation of a red button of erythrocytes in the bottom of the wells was considered an inhibition of agglutination.

## Immunofluorescence staining and microscopy

When confocal microscopy was planned, cells were cultured in 8-well chamber slides (#1.5) or Ibidi 96-well plates with a confocal bottom (#1.5). Cells were infected with BKPyV (MOI 200) or antibody-neutralized BKPyV and fixed at 6 hpi for 20 minutes by addition of 16% paraformaldehyde (PFA) at 37°C to achieve a final concentration of 4% PFA. For immunofluorescence staining, permeabilization and blocking were performed for 1 hour at room temperature in Dulbecco's phosphate-buffered saline (DPBS) with 0.3% Triton X100 and 5% goat serum. Primary antibody staining was done overnight at 4°C, while secondary antibody staining was done for 1 hour at room temperature the following day. The following antibodies were used to stain cellular proteins: mouse monoclonal anti-Rab7 (sc-376362, 1:50, SCBT), mouse monoclonal anti-EEA1 (sc-365652, 1:50, SCBT), and mouse monoclonal anti-Bap31 (Abcam 112993). The following antibodies were used to stain BKPyV proteins: mouse monoclonal anti-SV40LTag (Pab416, 1:100, Merck Millipore), mouse monoclonal anti-Vp1 (Virostat 4942, 2.8 μg/ml) and rabbit serums against BKPyV agnoprotein (1:1000) [91,92] and BKPyV Vp1 (1:1000) [93]. When the mouse monoclonal anti-Vp1 (Virostat 4942) was used in combination with mouse monoclonal anti-EEA1 (sc-365652), the Vp1 antibody was directly labeled with CF594 according to the manufacturer's instructions using the Mix-n-Stain CF Dye Antibody Labeling Kit (Biotium, cat#92448). Secondary antibodies used were goat anti-rabbit Alexa Fluor 488, anti-mouse Alexa Fluor 568, and anti-mouse-IgG2A or IgG1 with Alexa Fluor 568 or 647, all from Invitrogen. Nuclear staining was done with 1x DAPI or 1x Draq5 for 5–10 minutes. Confocal images and Z-stacks were acquired using an LSM800 confocal microscope with the Zeiss Zen blue software and a 40x water objective (NA 1.2) or 63x oil objective (NA 1.4). The number of Bap31-foci from immunofluorescent images was quantitated using the CellProfiler software version 4.2.6 [94].

## Transfections and plasmids

Lipofectamine 3000 was used to transfect RPTECs according to the manufacturer's instructions. Transfections were performed in 8-well chamber slides or Ibidi 96-well plates with 250 nanograms of DNA, 0.75 μl Lipofectamine 3000, and 0.5 μl P3000 per well.

The following plasmids were used: mCherry-Rab5, mCherry-Rab7a, pLAMP1-mCherry, EGFP-LAMP1, and pmTurquoise2-ER. mCherry-Rab5 (Addgene plasmid #49201) [95] and mCherry-Rab7A (Addgene plasmid #61804) [96] was a gift from Gia Voeltz. PmTurquoise2-ER (Addgene #36204) [97] and pLAMP1-mCherry (Addgene #45147) [98] were gifts from Dorus Gadella and Amy Palmer, respectively.

## Vacuole imaging, time-lapse microscopy, and live-cell imaging with organelle markers

Cells were examined for vacuolization with phase-contrast microscopy using a Nikon TE2000-microscope with a 10x (NA 0.13) and a 20x objective (NA 0.45) and the NIS Elements Basic Research software or oblique contrast microscopy using an automated CellDiscoverer 7 widefield microscope with a 20x objective (NA 0.8) and the Zeiss Zen blue software. Quantitation of vacuolized cells was done in ImageJ/Fiji. Vacuolized cells were manually counted and the percentage of vacuolized cells was calculated by dividing the number of vacuolized cells by the total number of nuclei as shown by Hoechst-staining.

For time-lapse microscopy, mock-infected and BKPyV-infected RPTECs were cultured in Ibidi 96-well plates with confocal glass bottom in REGM with CellTox dye (1x). Images were acquired every 30 minutes using an automated Zeiss CellDiscoverer 7 microscope with a 20x objective (NA 0.95) at $37°C$ with humidity and 5% $CO_2$. When neutralizing antibodies were used, they were added before the start of the time-lapse.

Markers for organelles and endo-/lysosomes were used for live-cell confocal and widefield imaging. RPTECs were cultured in 8-well chamber slides (#1.5) or Ibidi 96-well plates with a confocal bottom (#1.5). To load cells with Texas Red conjugated dextran kDa 70 (Thermo Fisher), cells were pulsed for at least 2 hours or overnight with Texas Red conjugated dextran 70 kDa (0.5 mg/ml) before BKPyV infection, washing, and image acquisition. For dextran loading of infected RPTECs, the dextran was added to cells at least 4 hours before imaging. To visualize fluorescently tagged Rab5, Rab7, Lamp1, and ER-marker, RPTECs were first transfected, then infected and imaged. For imaging of early-wave vacuolization, RPTECs were infected 2–3 days post-transfection (dpt) and imaged 5–8 hpi. For late-wave vacuolization, cells were infected 1 dpt and imaged at 4 dpi/5 dpt. Lysoview 488 and 633 (Biotium), Mitoview 633 (Biotium), and Hoechst dye (Invitrogen) were used for live-cell imaging according to the manufacturer's instructions. Lysoview (1:1000), Mitoview (25 nM), and Hoechst dye (0.1–0.5 ug/ml) were added to cells 30 to 60 minutes before imaging. All images were acquired using an automated Zeiss CellDiscoverer 7 widefield microscope with a 50x water objective (NA 1.2) or a Zeiss LSM800 confocal microscope with a 40x water objective (NA 1.2). Cells were imaged at $37°C$ with humidity and 5% $CO_2$.

## Dot blot

One µl of serial dilutions of gradient-purified BKPyV that had been heat-treated at $100°C$ for 10 minutes, fixed in 4% PFA for 10 minutes, or untreated (native) was applied to a nitrocellulose membrane. The membrane was first blocked using Odyssey LI-COR blocking buffer for 1 hour followed by incubation with primary antibodies, rabbit Vp1 antiserum, and mouse monoclonal Vp1 antibody (Virostat; 4942) overnight. Lastly, the membrane was incubated with secondary antibodies, goat anti-rabbit 800CW, and goat anti-mouse 680RD (LI-COR Biosciences), all for one hour. Detection was done with the Odyssey CLx imaging studio and Image Studio software.

## Transmission electron microscopy

The protocol was adapted from two reports [99, 100]. The samples were first fixed in 0.5% glutaraldehyde and 4% formaldehyde in PHEM-buffer (60 mM PIPES, 25 mM HEPES, 10 mM EGTA, 4 mM $MgSO_4·7H_2O$) and then fixed with 4% formaldehyde, 0.5% glutaraldehyde, and 0.05% malachite green in PHEM-buffer using a Ted Pella microwave processor. Next samples were post-fixed with osmium-reduced ferrocyanide (1% osmium tetroxide, 1% $K_3Fe(CN)_6$ in

0.1 M cacodylic acid buffer), treated with 1% tannic acid and 1% uranyl acetate, dehydrated in increasing concentrations of ethanol and embedded in an Epon-equivalent. A Leica Ultracut with a Diatome diamond knife was used to cut 70 nm sections, which were imaged using a Hitachi HT7800 transmission electron microscope with a Xarosa camera.

### Quantitative PCR

To monitor progeny release, supernatant BKPyV DNA was quantitated using a BKPyV-specific quantitative PCR [101]. Supernatants were harvested at 24, 48, and 72 hpi, and the 24 hpi time point was used as input.

### Inhibitors

The following inhibitors were used: EHT1864 (25 μM), SP600125 (10 μM), ethylisopropyla-miloride (EIPA; 10 μM) and bafilomycin A1 (20 and 200 nM). All inhibitors were from Med-ChemExpress and were diluted in DMSO. DMSO was used as a drug control. To assess their effect on vacuolization, RPTECs were either pretreated before BKPyV infection (MOI 200) or the cells were treated from 0 hpi or 2 hpi. Vacuolization was examined during the first 24 hpi by widefield microscopy. In the vacuolization assay, cells were infected (MOI 200) followed by drug treatment from 2 hpi and widefield microscopy at 6 hpi. The percentage of vacuolized cells was determined by dividing the number of vacuolized cells by the total number of cells as determined by Hoechst staining.

To quantify BKPyV replication, RPTECs were infected and fixed in methanol for ten minutes at 48 hpi before immunofluorescence staining for LTag and Vp1, or agnoprotein was performed as previously described [87]. The number of infected cells was then quantitated with ImageJ/Fiji [102] or Cellprofiler [94].

### Supporting information

**S1 Fig. Phase-contrast microscopy of BKPyV-infected RPTECs from different donors reveals early and late vacuolization.** (A) Widefield microscopy with oblique contrast of BKPyV-infected RPTECs at 6 hpi (MOI 0.1, 1.0, 10, 200 and 1000). Images are representative images from Fig 1C. (B) Widefield microscopy of RPTECs from Sciencell infected with BKPyV (MOI 5 or 100) at 12 and 96 hpi. Representative images from a single experiment are shown.
(TIF)

**S2 Fig. BKPyV-induced vacuolization is cell-type dependent.** Phase-contrast images of Mock-infected and BKPyV-infected cells (MOI 100) at 24 hpi and 96 hpi, (A) HeLa cells, and (B) CV-1 cells. Representative images from three independent experiments are shown. (C) Confocal microscopy following immunofluorescence staining at 6 hpi of BKPyV-infected HeLa cells, CV-1 cells, and RPTECs (MOI 200), respectively. A mouse monoclonal antibody against Rab7 (red) was combined with a mouse monoclonal antibody against BKPyV Vp1 (green). Nuclei are stained with Dapi (blue). Representative images from a single independent experiment are shown. Scale bar = 10 μm.
(TIF)

**S3 Fig. Characterization of the anti-BKPyV Vp1 activity of a mouse monoclonal antibody.** (A) Hemagglutination inhibition assay with BKPyV after pre-incubation with a mouse mono-clonal antibody against BKPyV Vp1 (Vp1 mAb, Virostat 4942). A negative control without BKPyV and a positive control without the Vp1 mAb are included. Hemagglutination was fully inhibited with a 100–6400 times dilution and partly inhibited with a 12 800–51 200 times

dilution. A representative image from two independent experiments is shown. (B) Dot blot of native (untreated), heat-treated, and PFA-fixed gradient-purified BKPyV. After treatment, 1 μl virus was applied to a nitrocellulose membrane before the membrane was blocked, stained with primary and secondary antibodies, and imaged with a LI-COR Imaging system. As primary antibodies, a rabbit BKPyV Vp1 antiserum and the mouse monoclonal antibody against BKPyV Vp1 (Virostat 4942) were used, respectively.
(TIF)

**S4 Fig. BKVLPs induce early-wave but not late-wave vacuolization, and the early vacuolization is transient.** Phase-contrast images of RPTECs at 24 hpi and 96 hpi with only BKVLPs (equal to MOI 200 based on the hemagglutination titer), a combination of BKVLPs (equal to MOI 200) and BKPyV (MOI 1), or a combination of BKVLPs (equal to MOI 1) and BKPyV (MOI 1). Representative images from two independent experiments are shown.
(TIF)

**S5 Fig. Early vacuoles are endo-/lysosomal structures.** Live-cell confocal microscopy of mock-infected and BKPyV-infected RPTECs (MOI 100–200) with markers of endo-/lysosomes and the ER at 6 hpi. (A) RPTECs were incubated overnight with Texas Red conjugated dextran (red) and stained with Lysoview 488 (green). Scale bar = 5 μm. (B) RPTECs that transiently express Rab7-mCherry (red), a marker of late endosomes and lysosomes, and Lamp1 (green), a marker of lysosomes. Scale bar = 10 μm. (C) RPTECs that transiently express Rab7-mCherry (red) and mTurquoise2-ER (green). Scale bar = 10 μm. Representative images from two independent experiments are shown.
(TIF)

**S6 Fig. Late vacuoles are endo-/lysosomal structures.** Live-cell oblique contrast and fluorescence widefield microscopy of mock-infected and BKPyV-infected RPTECs (MOI 1) with markers of endo-/lysosomes and the ER at 96 hpi. (A) RPTECs pulsed with Texas Red conjugated dextran (red) for 4 hours. (B) Transient expression of Rab7-mCherry (red). (C) Transient expression of Lamp1-mCherry (red). D) Transient expression of mTurquoise2-ER (green) and Rab7-mCherry (red). Representative images from two independent experiments are shown. Scale bar = 5 μm for all images.
(TIF)

**S7 Fig. BKPyV Vp1 is associated with early endosomes, mainly in the form of native virions.** Confocal microscopy of mock-infected and BKPyV-infected RPTECs (MOI 200) at 1 hpi following immunofluorescence staining. A mouse monoclonal antibody against early endosomal protein early endosome antigen 1 (EEA1) (red) was used in combination with the rabbit BKPyV Vp1 antiserum (green) and the mouse monoclonal antibody against BKPyV Vp1 (grey). Nuclei are stained with DAPI (blue). Scale bar = 5 μm. Representative images from two independent experiments are shown.
(TIF)

**S8 Fig. Transient vacuolization mainly occurs in recently infected RPTECs surrounding lysed cells.** (A) Representative images from time-lapse microscopy of BKPyV-infected RPTECs (MOI 0.1) at 78.5 and 90 hpi. Cells were imaged with widefield microscopy, and dead cells were stained with CellTox (green). Scale bar = 50 μm. (B) Time-lapse microscopy of BKPyV-infected RPTECs (MOI 0.1) demonstrating two examples of transient vacuolization (red boxes). Cells were imaged with widefield microscopy, and dead cells were stained with CellTox (green). Scale bar = 20 μm. (C) Immunofluorescence staining against Vp1 (green) and LTag (red) in BKPyV-infected RPTECs (MOI 1) at 72 hpi from Fig 10B. Note RPTECs with

strong cytoplasmic Vp1-staining without nuclear staining of LTag and Vp1. Nuclei were stained with DAPI. Scale bar = 10 μm.
(TIF)

**S9 Fig. A BKPyV-specific neutralizing antibody inhibits vacuolization.** Oblique contrast microscopy of BKPyV-infected RPTECs (MOI 1) at 72 hpi. The cells were untreated or treated with either a BKPyV-Vp1 specific neutralizing antibody or a control antibody from 24 hpi. Representative images from Fig 10C and 10D are shown. Scale bar = 50 μm.
(TIF)

**S10 Fig. Effect of EHT1864, SP600125, and EIPA on BKPyV replication and vacuolization.** Immunofluorescence staining of LTag (red) and agnoprotein (green) in BKPyV-infected RPTECs (MOI 5) at 48 hpi after treatment 2–48 hpi with (A) EHT1864 at 25 μM and (B) Sp600125 at 10 μM. Cells with LTag expression were counted, and data was presented as normalized LTag expression (%) compared to the DMSO control. Error bars = SD and n = 2 except for Sp600125 where n = 1. (C) Immunofluorescence staining of Vp1 (red) and agnoprotein (green) in BKPyV-infected RPTECs (MOI 1) at 48 hpi after treatmentwith EHT1864 at 25 μM 24–48 hpi. Representative images from two independent experiments are shown. (D) Oblique contrast microscopy of BKPyV-infected RPTECs (MOI 1) at 96 hpi after treatment with EHT1864 at 25 μM24–96 hpi. Representative images from three individual experiments were shown. (E) Immunofluorescence staining of LTag (red) and agnoprotein (green) in BKPyV-infected RPTECs (MOI 5) at 48 hpi after treatment with EIPA at 10 μM 2–48 hpi. Cells with LTag expression were counted, and data was presented as normalized LTag expression (%) compared to the DMSO control. Error bars = SD and n = 2. (F) Oblique contrast microscopy of BKPyV-infected RPTECs (MOI 100) at 24 hpi. Cells were pretreated overnight with EHT1864 at 25 μM or EIPA at 10 μM and up to imaging at 24 hpi. Representative images from one single experiment are shown. For all experiments DMSO was used as a drug control.
(TIF)

**S11 Fig. Effect of bafilomycin A on BKPyV replication.** Immunofluorescence staining of DMSO or Bafilomycin A treated BKPyV (MOI 5) infected RPTECs at 48 hpi, using antibodies against LTag (red) and Vp1 (green). Cells were bafilomycin A treated from (A) 2–48 hpi, (B) 0–2 hpi, or (C) 0–48 hpi. Nuclei are stained with DAPI (blue). Representative images from two independent experiments are shown. D) Quantification of LTag expression in (A). Data is presented as LTag expression normalized to the DMSO control. Error bars represent SD and n = 2.
(TIF)

**S1 Video. Time-lapse microscopy of RPTECs infected with BKPyV MOI 200 from 3 to 96.5 hpi.** Scalebar = 50 μm. An image was acquired every 30 minutes.
(AVI)

**S2 Video. Time-lapse microscopy of RPTECs infected with BKPyV MOI 10 from 3 to 96.5 hpi.** Scalebar = 50 μm. An image was acquired every 30 minutes.
(AVI)

**S3 Video. Time-lapse microscopy of RPTECs infected with BKPyV MOI 1 from 3 to 96.5 hpi.** Scalebar = 50 μm. An image was acquired every 30 minutes.
(AVI)

**S4 Video. Time-lapse microscopy of RPTECs infected with BKPyV MOI 0.1 from 3 to 96.5 hpi.** Scalebar = 50 μm. An image was acquired every 30 minutes.
(AVI)

**S5 Video. Time-lapse microscopy of mock-infected RPTECs from 3 to 96.5 hpi.**
Scalebar = 50 μm. An image was acquired every 30 minutes.
(AVI)

**S6 Video. Time-lapse microscopy of RPTECs infected with BKPyV MOI 0.1 from 52 to 82 hpi.** A BKPyV-specific neutralizing antibody (left window) or no antibody was added (right window). An image was acquired every 30 minutes.
(AVI)

**S1 Data. Data used to generate graphs in Figs 1, 8, 10, 11, S10 and S11.**
(XLSX)

## Acknowledgments

We are grateful to the Advanced Microscopy Core Facility (AMCF) at the Institute of Medical Biology, UiT–The Arctic University of Tromsø for providing access to instruments and to Kenneth Bowitz Larsen (AMCF) and Randi Larsen (AMCF) for processing samples for electron microscopy. We thank Garth D. Tylden (University Hospital of North Norway) and Kristian Prydz (University of Oslo) for helpful discussions and critical manuscript reading. We are grateful to Hans H. Hirsch (University of Basel) for essential discussions and for providing BKVLPs.

## Author Contributions

**Conceptualization:** Elias Myrvoll Lorentzen, Stian Henriksen, Christine Hanssen Rinaldo.

**Data curation:** Elias Myrvoll Lorentzen, Stian Henriksen.

**Formal analysis:** Elias Myrvoll Lorentzen, Stian Henriksen.

**Funding acquisition:** Elias Myrvoll Lorentzen, Christine Hanssen Rinaldo.

**Investigation:** Elias Myrvoll Lorentzen, Stian Henriksen.

**Methodology:** Elias Myrvoll Lorentzen, Stian Henriksen.

**Project administration:** Christine Hanssen Rinaldo.

**Supervision:** Christine Hanssen Rinaldo.

**Visualization:** Elias Myrvoll Lorentzen, Stian Henriksen, Christine Hanssen Rinaldo.

**Writing – original draft:** Elias Myrvoll Lorentzen.

**Writing – review & editing:** Elias Myrvoll Lorentzen, Stian Henriksen, Christine Hanssen Rinaldo.

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
