## [Decision Letter · Decision Letter 0]

30 Aug 2024

Dear Professor Rinaldo,

Thank you very much for submitting your manuscript "Massive entry of BK Polyomavirus induces transient cytoplasmic vacuolization of human renal proximal tubule epithelial cells" for consideration at PLOS Pathogens. As with all papers reviewed by the journal, your manuscript was reviewed by members of the editorial board and by several independent reviewers. The reviewers appreciated the attention to an important topic. Based on the reviews, we are likely to accept this manuscript for publication, providing that you modify the manuscript according to the review recommendations.

Sincerely,

Walter J. Atwood

Academic Editor

PLOS Pathogens

Robert Kalejta

Section Editor

PLOS Pathogens

Michael Malim

Editor-in-Chief

PLOS Pathogens

orcid.org/0000-0002-7699-2064

Reviewer Comments (if any, and for reference):

Reviewer's Responses to Questions

**Part I - Summary**

Reviewer #1: In this study, the Rinaldo group explores cytoplasmic vacuolization induced by BK polyomavirus in human renal proximal tubule epithelial cells. They show that infection at very high doses can induce rapid, transient vacuolization in both polarized and non-polarized RPTEC, whereas as low doses, vacuolization occurs late during infection, after progeny virus production and release. The vacuoles themselves appear to be enlarged endo/lysosomes, depend on virus binding to cell surface receptors, and their formation is blocked by bafilomycinA1, an inhibitor of endosomal acidification. Progeny virus release and cell lysis precede vacuole formation and do not appear dependent on it. Many of their findings parallel the SV40 system, where cell surface binding (and GM1 ganglioside engagement) triggers vacuolization, which can occur early or late upon virus replication, and binding of progeny virus induces vacuole formation late. But there are some interesting differences, including (presumably) the receptor involved, the responsible signaling pathways (at least in the RPTEC, which were not studied for SV40), the presence or absence of virions in the vacuoles, and the relative timing of vacuolization and virus release. Overall, this work appears carefully done and most of the conclusions justified (although see point three below), but the work is largely descriptive and somewhat incremental.

Specific comments

1. The neutralizing antibody clearly inhibits vacuolization, but it might have other effects on the virus in addition to preventing receptor binding. They should be more circumspect in interpreting this experiment.

2. Is the high MOI required for early vacuolization likely to be achieved in a natural infection?

3. The conclusion that virus can undergo cell-to-cell spread (based on the focal clustering of cells containing late vacuoles around lysed cells) is interesting but should be better documented and justified. If cell-to-cell spread is important as stated in the abstract, late vacuole formation should be resistant to antibody neutralization, but they don’t see this in Figure 2D/E.

Reviewer #2: Cytoplasmic vacuolization of renal tubular cells is commonly observed in clinical histological samples, both in association with viral infections and in tubular injury of multiple otheetiologies. From the morphological point of view all forms of vacuolization are lumped together.

The current study provides significant new information that helps understand and better categorize BKPyV associated cellular vacuolization. The transient nature of vacuolization and its relationship to the infectious cell cycle are also important findings of this study.

The study is comprehensive and well-illustrated.

Reviewer #3: The manuscript from the Rinaldo group tests the idea whether BK polyomavirus, like SV40, causes vacuolisation of the cells it infects. They utilise the primary renal proximal tubular epithelial cells, which are the current best cell model for this virus and they find that BK can cause vacuole formation. They demonstrate that high tires of virus cause vacuole formation at both early and late time points, whereas lower titres only cause the phenotype at later time points, correlating with virus release. A series of analyses demonstrates both the requirements for vacuole formation and provides some information as to the cell biology of these vacuoles. The relevance of their formation in the virus lifecycle remains unclear. Overall the experiments are well performed and their are controls in the assays. A major sticking point relates to the very high titres of virus used in the assays, it could call into question the physiological relevance of the findings; however, lower titres can induce vacuole formation and there are reports in the literature of vacuole formation in kidney tissues from patients with BK-induced kidney disease.

**Part II – Major Issues: Key Experiments Required for Acceptance**

Reviewer #1: Item #3 above should be address substantively

Reviewer #2: The experimental aspects are well planned and well executed.

Reviewer #3: The authors could further test if this is truly dependent on GT1b/GD1b by either enzymatic inhibition of their production or siRNA/shRNA knockdown and performing the infection assays.

The authors could further demonstrate that replication is not needed by using VLPs in addition to the heat inactivated virus.

**Part III – Minor Issues: Editorial and Data Presentation Modifications**

Reviewer #1: 1. On line 152, they should emphasize that they used a low MOI to explore late vacuole formation dependent on virus replication and progeny virus production.

Reviewer #2: The authors demonstrated that "early" and "late" vacuolization represent an identical process related to the massive entry of BKPYV. I suggest that further qualification be used after early and late vacuolization (such as early or late wave of vacuolization, or early and late surge of vacuolization) indicating that the temporality is related to the time in the course of the study, rather than to the individual cell infectious cycle. In the manuscript the idea eventually becomes clear, however, until the results for the findings in the late vacuolization are presented, there is potential ambiguity regarding this issue.

Reviewer #3: The discussion needs to provide clearer thoughts on how BK induces vacuoles as presumably the virus induces productive infection in HeLa and CV1 cells but does not cause vacuole formation, hence the infection process per se does not seem sufficient for this (yet that is the suggestion in the RPTE cells). What is different between the cell types?

It would be very useful for the authors either in the methods or in the results to provide clearer explanation as to why different MOIs are used for different experiments (these can vary quite significantly between experiments in each figure).

PLOS authors have the option to publish the peer review history of their article (what does this mean?). If published, this will include your full peer review and any attached files.

Reviewer #1: No

Reviewer #2: **Yes: **Cinthia Drachenberg, M.D.

Reviewer #3: No

Figure Files:

Data Requirements:

Reproducibility:

References:

---

## [Editor Report · Decision Letter 1]

20 Oct 2024

Dear Professor Rinaldo,

We are pleased to inform you that your manuscript 'Massive entry of BK Polyomavirus induces transient cytoplasmic vacuolization of human renal proximal tubule epithelial cells' has been provisionally accepted for publication in PLOS Pathogens.

Best regards,

Walter J. Atwood

Academic Editor

PLOS Pathogens

Robert Kalejta

Section Editor

PLOS Pathogens

Michael Malim

Editor-in-Chief

PLOS Pathogens

orcid.org/0000-0002-7699-2064
---

## [Editor Report · Acceptance letter]

26 Oct 2024

Dear Professor Rinaldo,

We are delighted to inform you that your manuscript, " Massive entry of BK Polyomavirus induces transient cytoplasmic vacuolization of human renal proximal tubule epithelial cells ," has been formally accepted for publication in PLOS Pathogens.

Best regards,

Michael Malim

Editor-in-Chief

PLOS Pathogens

orcid.org/0000-0002-7699-2064